# Pathogen clonal expansion underlies multiorgan dissemination and organ-specific outcomes during murine systemic infection

Karthik Hullahalli[1,2]*, Matthew K Waldor[1,2]*

[1]Department of Microbiology, Harvard Medical School, Boston, United States; [2]Division of Infectious Diseases, Brigham & Women's Hospital, Boston, United States

**Abstract** The dissemination of pathogens through blood and their establishment within organs lead to severe clinical outcomes. However, the within-host dynamics that underlie pathogen spread to and clearance from systemic organs remain largely uncharacterized. In animal models of infection, the observed pathogen population results from the combined contributions of bacterial replication, persistence, death, and dissemination, each of which can vary across organs. Quantifying the contribution of each these processes is required to interpret and understand experimental phenotypes. Here, we leveraged STAMPR, a new barcoding framework, to investigate the population dynamics of extraintestinal pathogenic *Escherichia coli*, a common cause of bacteremia, during murine systemic infection. We show that while bacteria are largely cleared by most organs, organ-specific clearance failures are pervasive and result from dramatic expansions of clones representing less than 0.0001% of the inoculum. Clonal expansion underlies the variability in bacterial burden between animals, and stochastic dissemination of clones profoundly alters the pathogen population structure within organs. Despite variable pathogen expansion events, host bottlenecks are consistent yet highly sensitive to infection variables, including inoculum size and macrophage depletion. We adapted our barcoding methodology to facilitate multiplexed validation of bacterial fitness determinants identified with transposon mutagenesis and confirmed the importance of bacterial hexose metabolism and cell envelope homeostasis pathways for organ-specific pathogen survival. Collectively, our findings provide a comprehensive map of the population biology that underlies bacterial systemic infection and a framework for barcode-based high-resolution mapping of infection dynamics.

**\*For correspondence:**
Hullahalli@g.harvard.edu (KH);
mwaldor@research.bwh.harvard.edu (MKW)

**Competing interest:** The authors declare that no competing interests exist.

## Introduction

Sustained bacterial survival in the bloodstream and establishment within otherwise sterile organs is often fatal and severely burdens healthcare infrastructure (*Christaki and Giamarellos-Bourboulis, 2014*). In healthy individuals, bacteria frequently breach epithelial barriers and enter into the circulatory system, but are rapidly eliminated by complement and other humoral factors and/or captured and killed by liver- and spleen-resident phagocytic cells, clearing the infection and preventing sustained bacteremia (*Jenne and Kubes, 2013*; *Krenkel and Tacke, 2017*). However, some pathogenic bacteria can at least temporarily evade or resist these host defenses to potentially enable subsequent dissemination (*Smith et al., 2010*; *Armbruster et al., 2019*; *Holmes et al., 2021*; *Ercoli et al., 2018*; *Gresham et al., 2000*; *Siggins et al., 2020*). Extraintestinal pathogenic *Escherichia coli* (ExPEC), a set of pathogenic *E. coli* isolates that cause disease outside of the intestine, are the leading causes of bacteremia in humans (*Dale and Woodford, 2015*). In mouse models of systemic infection, ExPEC strains exhibit markedly delayed clearance from the liver and spleen compared to commensal *E.*

*coli*, suggesting that despite their eventual clearance, ExPEC possess strategies to persist longer within organs, thereby delaying the resolution of infection (*Smith et al., 2010*). However, when and where these clearance delays manifest is not fully understood. More broadly, the pathogen population dynamics of systemic infections is highly complex and not well characterized, and understanding the interplay between bacterial spread, replication, persistence, and clearance can help inform therapeutic interventions (*Abel et al., 2015b*).

A variety of immunological and microscopy approaches have been employed to investigate the host factors that regulate bacterial dissemination (*Ercoli et al., 2018*; *Siggins et al., 2020*; *Jorch et al., 2019*). For example, replication of *Streptococcus pneumoniae* within CD169[+] splenic macrophages (*Ercoli et al., 2018*) or *Staphylococcus aureus* within GATA6[+] peritoneal macrophages (*Jorch et al., 2019*) have been shown to enable systemic dissemination. However, such approaches require infection models with very large bacterial burdens and are not easily amenable to high-throughput or population- and animal-wide analyses. Furthermore, these methodologies require prior understanding of within-host pathogen population dynamics during infection, which then enable hypothesis generation to dissect underlying mechanisms of infection.

Several different bacterial population-level approaches have been employed to study infection dynamics (*Abel et al., 2015b*; *Fiebig et al., 2020*; *Grant et al., 2008*; *Blundell and Levy, 2014*), including systemic infection in *Pseudomonas aeruginosa, S. aureus,* and *S. pneumoniae* (*Bachta et al., 2020*; *Pollitt et al., 2018*; *Gerlini et al., 2014*). Most of these methodologies involve barcoding otherwise identical bacteria and enumerating the abundance of barcodes to follow lineages (*Blundell and Levy, 2014*; *Jasinska et al., 2020*). One approach known as STAMP (Sequence Tag-Based Analysis of Microbial Populations) leverages deep sequencing of bacteria barcoded with random DNA sequence tags and population genetics frameworks to quantify bottlenecks, dissemination patterns, and, more recently, replication rates (*Abel et al., 2015a*; *Mahmutovic et al., 2021*). STAMP was initially developed to study *Vibrio cholerae* intestinal colonization, and subsequently applied to *Listeria monocytogenes* dissemination from the intestine, *P. aeruginosa* bacteremia, and *S. pneumoniae* nasopharyngeal colonization (*Bachta et al., 2020*; *Zhang et al., 2021*; *Zhang et al., 2017*; *Liu et al., 2021*). We recently created STAMPR, a successor to STAMP that relies on a new computational framework and employs additional metrics to study infection population dynamics at higher resolution (*Hullahalli et al., 2021*).

Previous analyses of infection population dynamics have often been limited by either a small number of barcodes, organ samples, or time points. Here, we leveraged STAMPR to comprehensively map ExPEC population dynamics following dissemination in blood after intravenous (i.v.) inoculation using ~1100 barcodes across over 400 samples from 6 time points and 12 within-animal compartments. Intravenous bacteremia models vary in outcome depending on strain, dose, and length of time post inoculation (*Smith et al., 2010*; *Pollitt et al., 2018*; *Diabate et al., 2015*; *McAdow et al., 2011*). We adopted a nonlethal dose of ExPEC to investigate the delayed clearance of this pathogen and monitored the clearance phase to uncover the population dynamics that underlie delayed clearance. Although ExPEC is largely cleared by most organs, some organs, most notably the liver, stochastically fail to clear the pathogen. We show that these failures are attributable to the massive expansion of very few bacterial cells from the inoculum and explain the marked variation in bacterial burden observed between animals and organs. In certain instances, these clones disseminate and alter the population structure within distal organs. Despite these dramatic clonal expansion events, the magnitude of host bottlenecks remains consistent within organs between animals. However, bottleneck sizes are plastic and highly sensitive to experimental conditions. Both macrophage depletion and increased dose widen bottleneck sizes, but stochastic clonal expansion events can obscure colony forming unit (CFU)-based detection of these phenotypes. Finally, guided by our granular assessment of population dynamics, we identify ExPEC genes required for the establishment of and persistence through infection as well as clonal expansion; unexpectedly, bacterial hexose metabolism distinguishes these processes in an organ-specific manner. Collectively, our findings explain the population-level phenotypes that underpin ExPEC systemic dissemination and deepen our understanding of the origins of within-host bacterial populations.

## Results

### Experimental design and analyses

A key metric for understanding within-host population dynamics is the founding population size (FP) (*Abel et al., 2015a*; *Mahmutovic et al., 2021*). FP quantifies the number of unique cells from the inoculum that give rise to the pathogen population at a sampling point. Higher FP values are indicative of wider (more permissive) bottlenecks. FP can be measured by an equation derived by *Krimbas and Tsakas, 1971*, the output of which is referred to as $N_b$. The greater the difference in barcode frequencies between input and output samples, the smaller the $N_b$. In STAMPR, an algorithmic adjustment is made to $N_b$ that limits the influence of disproportionately abundant barcodes, resulting in a new metric called $N_r$ (*Hullahalli et al., 2021*, mSystems). The degree to which $N_b$ is biased by these 'outliers' can be exploited by calculation of the $N_r/N_b$ ratio, providing a measure of uniformity of barcode frequencies; this ratio is larger in samples where a few bacterial clones have markedly expanded.

We introduced ~1150 20 nt random and unique sequence barcodes into the ExPEC strain CFT073 (rpoS$^+$) (*Welch et al., 2002*; *Ristow and Welch, 2016*; *Hryckowian and Welch, 2013*) *lacZ* locus. The barcodes were stably maintained in the absence of selection for at least ~50 generations (*Figure 1—figure supplement 1A*) and did not modify the growth rate of the strains in lysogeny broth (LB) media compared to the parental strain (*Figure 1—figure supplement 1B*). Deep sequencing of the barcodes at different known bottleneck sizes (i.e., plated CFU) revealed that $N_r$ and $N_b$ both accurately quantified in vitro bottlenecks up to $10^5$ CFU for this library, confirming the fidelity of these metrics and the absence of substantial technical artifacts during library preparation (e.g., PCR jackpots, contaminating barcodes; *Figure 1—figure supplement 1C*).

Mice were injected intravenously with $10^7$ CFU of the barcoded ExPEC library and were sacrificed beginning 4 hr post infection (hpi) and then daily up to 5 days post infection (dpi) (*Figure 1A*). The blood, bile, left and right halves of the spleen, left and right kidney, left and right lung, and all four lobes of the liver were harvested at each time point. 4 mice each day were sacrificed from 0 to 4 days post infection, and 12 mice were sacrificed at 5 dpi, collectively yielding 384 CFU measurements across 12 suborgans and 6 time points. Sequencing for STAMPR calculations was carried out on 322 samples that had nonzero CFU. Organs containing as few as 1 CFU were sequenced and analyzed because such samples served as important within-run sequencing controls, yielding information about the level of noise due to technical artifacts (such as index hopping), as they should only possess one highly dominant barcode.

### Bacteria fail to be cleared in a fraction of animals

We first address the patterns of bacterial burden up to 5 dpi across the 12 compartments. As observed in previous studies (*Smith et al., 2010*), the highest CFU values were found in blood and most organs 4 hpi (0 dpi) and a few bacteria remained detectable in blood after 1 dpi (*Figure 1F*, *Figure 1—figure supplement 2*). The kidneys displayed a similar trend, although some bacteria were detected up to 5 dpi in some animals (*Figure 1F*, *Figure 1—figure supplement 3*). The spleen contained more bacteria than the kidneys but still trended towards clearance (*Figure 1F*, *Figure 1—figure supplement 2*). In the lung, however, even at 5 dpi, several hundred bacteria were present, suggesting that the lung has lower clearance capacities compared to the other organs (*Figure 1F*, *Figure 1—figure supplement 3*). The rapid pathogen elimination from blood likely reflects clearance by filtrative organs like the liver and spleen, which had the highest bacterial burdens 4 hpi, rather than killing by blood (e.g., complement), since mouse blood is permissive for *E. coli* growth (*Marcus et al., 1954*).

Clearance failures were most prominent in the liver, which yielded the highest CFU values 5 dpi (*Figure 1F*, *Figure 1—figure supplement 4*). Hepatic bacterial burdens were bimodal, with some mice containing ~$10^6$ bacterial cells while littermates had ~$10^4$-fold fewer bacteria. Notably, animals with high CFU had visible abscesses in the liver. Lobes of the liver that lacked visible abscesses did not have high CFU, arguing against the presence of occult abscesses. While CFU were usually cleared from organs outside the liver, high CFU were also occasionally observed in additional organs; for example, in one animal, the bacterial burden in the left and right lungs differed by 100-fold (*Figure 1—figure supplement 3*). Unlike the liver, the organs with elevated CFU did not harbor visibly apparent abscesses. We also observed three animals with very high bacterial burdens in the bile (~$10^7$ CFU/ml) aspirated from gallbladder samples from 4 and 5 dpi (*Figure 1—figure supplement 2*). Taken together, measurements of bacterial burden alone suggest a model whereby the pathogen distribution early

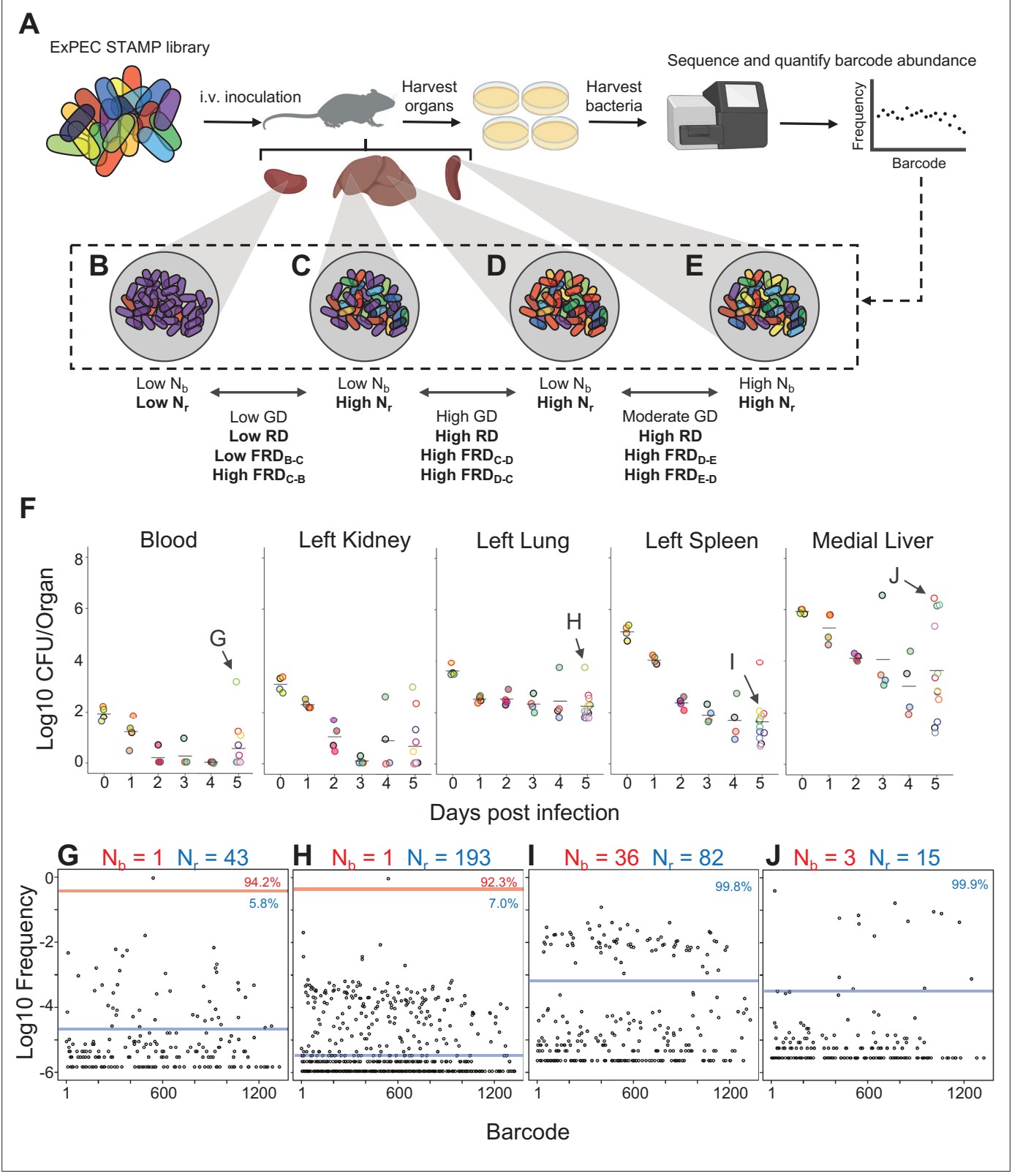

**Figure 1.** Experimental design and metrics and CFU dynamics. (**A**) The experimental schematic depicts the extraintestinal pathogenic *Escherichia coli* (ExPEC) barcoded library as colored bacteria, where colors indicate unique barcodes; the library was injected intravenously (i.v.) into mice, after which organs were harvested, homogenized, and plated for enumeration of CFU. Bacterial cells were scraped, pooled, and the barcode abundances in the output populations were quantified. The distributions of the barcodes (far-right graph) define the population structure of the organ, schematized in

*Figure 1 continued on next page*

*Figure 1 continued*

(**B–E**). (**B–E**) Graphical depiction of populations with different combinations of $N_b$ and $N_r$ values. These populations arise from (**B**) a tight bottleneck and subsequent expansion of purple cells, (**C, D**) a wider bottleneck and expansion of purple (**C**) or red (**D**) cells and (**E**) a wide bottleneck and even growth of all cells. Metrics used for comparisons between samples (genetic distance [GD], "resilient" genetic distance [RD], and fractional RD [FRD]) are indicated. (**F**) CFU recovered from select organs on days 0–5; the full CFU data set is provided in *Figure 1—figure supplements 2–4*. Points with the same fill and border color were obtained from the same animal. Arrows pointing to G–J correspond to the points where complete barcode frequency distributions are shown below (in **G–J**). (**G–J**) Barcode distributions and $N_b$ and $N_r$ values are shown. Red lines separate populations that were distinguished by the $N_r$ algorithm (*Hullahalli et al., 2021*). Blue line indicates the algorithm-defined threshold for noise. Percentages represent the relative abundance of barcodes within each region. **G and H** represent examples with highly abundant barcodes that skew $N_b$ values lower. (**G**) reproduced from Figure S5B (*Hullahalli et al., 2021*).

The online version of this article includes the following figure supplement(s) for figure 1:

**Figure supplement 1.** Barcode stability and creation of the STAMPR standard curve.

**Figure supplement 2.** Longitudinal CFU, $N_b$, and $N_r$ measurements from the blood, bile, and spleen.

**Figure supplement 3.** Longitudinal CFU, $N_b$, and $N_r$ measurements from the lungs and kidneys.

**Figure supplement 4.** Longitudinal CFU, $N_b$, and $N_r$ measurements from the liver.

during infection is highly predictable; however, by 5 dpi there is substantial variability in burden, and the range of this variability is organ-specific. To generalize, we can identify at least two trajectories from CFU alone. The first is successful clearance, where very few bacteria survive or persist till 5 dpi with a general decrease in CFU over time. The second appears to be variable organ-specific clearance failure, the most dramatic of which result in liver abscesses but were also detected in 'outlier' animals at least once in all organs.

## Clearance failures are mostly driven by clonal expansion

To determine the ExPEC population structure within organs, barcode frequencies were determined and $N_b$ and $N_r$ were calculated from the 322 samples that had nonzero CFU (*Figure 1G–J*, *Figure 1— figure supplements 2–4*). *Figure 1G–J* demonstrates examples of $N_b$ and $N_r$ values calculated from barcode frequency distributions. The distributions themselves provide intuitive snapshots of the bacterial population structure in single animals. *Figure 1G* demonstrates an animal with unusually high CFU in blood, where ~95% of reads corresponded to a single tag. The same barcode was highly abundant in the left lung of the same animal (*Figure 1H*), accounting for more than 90% of reads. In addition, this sample had an underlying population that comprised 7% of reads. Therefore, the elevated bacterial burden in the left lung of this animal results from a clone that was also circulating in blood. *Figure 1I* depicts bacteria from a spleen sample from a different animal than *Figure 1G* in which ~80 barcodes represented ~100 CFU, revealing that there was very little apparent replication (low $N_r/N_b$ ratio), similar to *Figure 1E*. In stark contrast, *Figure 1J* represents a liver lobe with an abscess from a different animal than *Figure 1I* with >$10^6$ CFU derived from only ~10 tags, an example of dramatic replication of very few cells.

The patterns described above (exemplified by *Figure 1B–E and G–J*) can be distilled by comparisons of $N_b$, $N_r$, and CFU values. For example, when $N_b$ and $N_r$ are similar to each other and to CFU, we can deduce that the bacterial population has undergone very little replication (e.g., *Figure 1I*). In contrast, organs with high CFU and low $N_r$ and $N_b$ values contain populations that resulted from expansion of very few cells (e.g., *Figure 1J*, *Figure 1—figure supplements 2–4*). At 4 hpi, when mice were still bacteremic, we observed that the number of blood CFU (~100) was nearly the same as the corresponding $N_r$ (~80), indicating that at this point most bacteria possessed different tags, suggesting that there has been little replication in blood following inoculation (*Figure 1—figure supplement 2*). Like the blood, at early time points both kidneys had $N_r$ values similar to CFU, indicating minimal net replication (*Figure 1—figure supplement 3*). Our approach also permits analysis of the underlying populations in animals that appear to be 'outliers' from others in the cohort. One example is an animal from 3 dpi where the left kidney possessed 0 CFU and the right had >10,000 CFU (*Figure 2—figure supplement 1A*). The population in the right kidney was represented by only eight tags, revealing that approximately eight bacterial cells from the inoculum had dramatically expanded (*Figure 2— figure supplement 1B*). In the spleen at 4 hpi (0 dpi), $N_r$ and $N_b$ values were ~$10^4$, slightly less than CFU values (*Figure 1—figure supplement 2*). By 5 dpi, the CFU recovered (~20) was very close to values of $N_r$, indicating that these remaining cells had not undergone substantial replication. One

notable exception was a sample from 5 dpi (mouse 23) that had a much higher bacterial burden than other mice (*Figure 2A*). Since this animal also had a markedly elevated $N_r$ (wider bottleneck) as well as an elevated $N_r/N_b$ (expanded clones over a diverse population), the increased bacterial burden can be attributed both to clonal expansion as well as a wider bottleneck, which is consistent with visual observations of barcode frequency distributions (*Figure 2B and C*). In the lungs, CFU approximated $N_r$, indicating that the bacterial population had undergone very little replication (*Figure 1—figure supplement 3*). This is consistent with the longitudinal CFU measurements, which plateaued by 3 dpi (*Figure 1F*).

As discussed above, some of the largest bacterial burdens in these experiments were present in the liver at 5 dpi (*Figure 1F*, *Figure 1—figure supplement 4*). The elevated hepatic CFU values were associated with clonal expansion events since $N_r$ and $N_b$ were orders of magnitude lower than CFU in animals with abscesses (*Figure 1J*, *Figure 1—figure supplement 4*). Even at 1 dpi, two animals had clonal expansion ($N_r/N_b$ >1000) in multiple lobes (*Figure 2E and G*, *Figure 1—figure supplement 4*) where ~ 1% of barcodes comprised a large majority of the reads (*Figure 2E–H*). Nevertheless, at 5 dpi all animals had similar $N_r$ values, despite some having abscesses and the concomitant 10,000-fold increase in bacterial burden (*Figure 1—figure supplement 4*). Thus, the immune and physical factors that govern bottlenecks appear to be highly consistent between animals. Instead, the focal and stochastic expansion of a few clones within abscesses, rather than a general increase in bacterial survival, explains the wide variance in liver CFUs. In general, these analyses revealed that $N_r$ values were more consistent and lower than CFU across all organs particularly at 5 dpi, revealing that most observations of very high bacterial burden are driven by very few bacterial clones (*Figure 1—figure supplements 2–4*). Apart from the lung, FP sizes decreased during the 5 days of observation. Therefore, the host forces that restrict the bacterial population in this model act throughout the infection and are not limited to the initial establishment of the pathogen population. However, each organ is associated with a specific and distinct probability of yielding clonal expansion, likely reflecting their underlying biological/compositional differences.

## Patterns of inter- and intra-organ dissemination

Comparisons of barcode frequencies between two organ samples can be used to quantify the relatedness between two pathogen populations in the host, effectively revealing inter- and intra-organ spread. This metric, known as genetic distance (GD), is small when two samples are highly related and large when samples are not related. Low GD values (~0.1–0.2) require that the same barcodes be relatively abundant, not simply present, in both samples. Spreading event(s) therefore must have occurred prior to sampling to allow for sufficient time to result in the same barcode(s) being highly abundant in both samples or be substantial enough to where a large number of bacteria are transferred. GD is also influenced by outliers, and so a different metric, called RD ('resilient' genetic distance), quantifies the number of barcodes that contribute to the similarity between two populations in a manner that takes into account all detected barcodes (*Hullahalli et al., 2021*; *Figure 1A*). For example, if both RD and GD are low, only a few barcodes contribute to the genetic similarity between samples. In contrast, if RD is high and GD is low, the samples are similar because they share many barcodes, which often occurs when samples closely resemble the inoculum through recent temporal separation. Therefore, RD provides a metric to distinguish between samples that are similar because they share few (low RD) or many (high RD) barcodes. RD can be normalized to the number of barcodes in a sample, and this 'fractional' RD (FRD) is a directional metric that quantifies the extent to which shared barcodes are represented by all barcodes in the sample. $FRD_{A-B}$ is low when the shared barcodes in samples A and B represent a small fraction of the total barcodes in sample B, and large when the shared barcodes represent a large fraction of the total barcodes in sample B. $FRD_{B-A}$ is the reverse, where the shared barcodes are normalized to the total barcodes in sample A (*Figure 1A*).

To depict dissemination patterns across the entire animal, we display GD and FRD values as heatmaps, which permit rapid visualization of the degree to which samples are similar (*Figure 3—figure supplements 1–6*). At 4 hpi, all organs possess highly similar bacterial populations that share many barcodes (*Figure 3—figure supplement 1*) as they have only recently been separated from the inoculum. After this initial capture of bacteria, GD between organs generally increases over time, suggesting that the populations are not substantially disseminating and mixing, and that populations

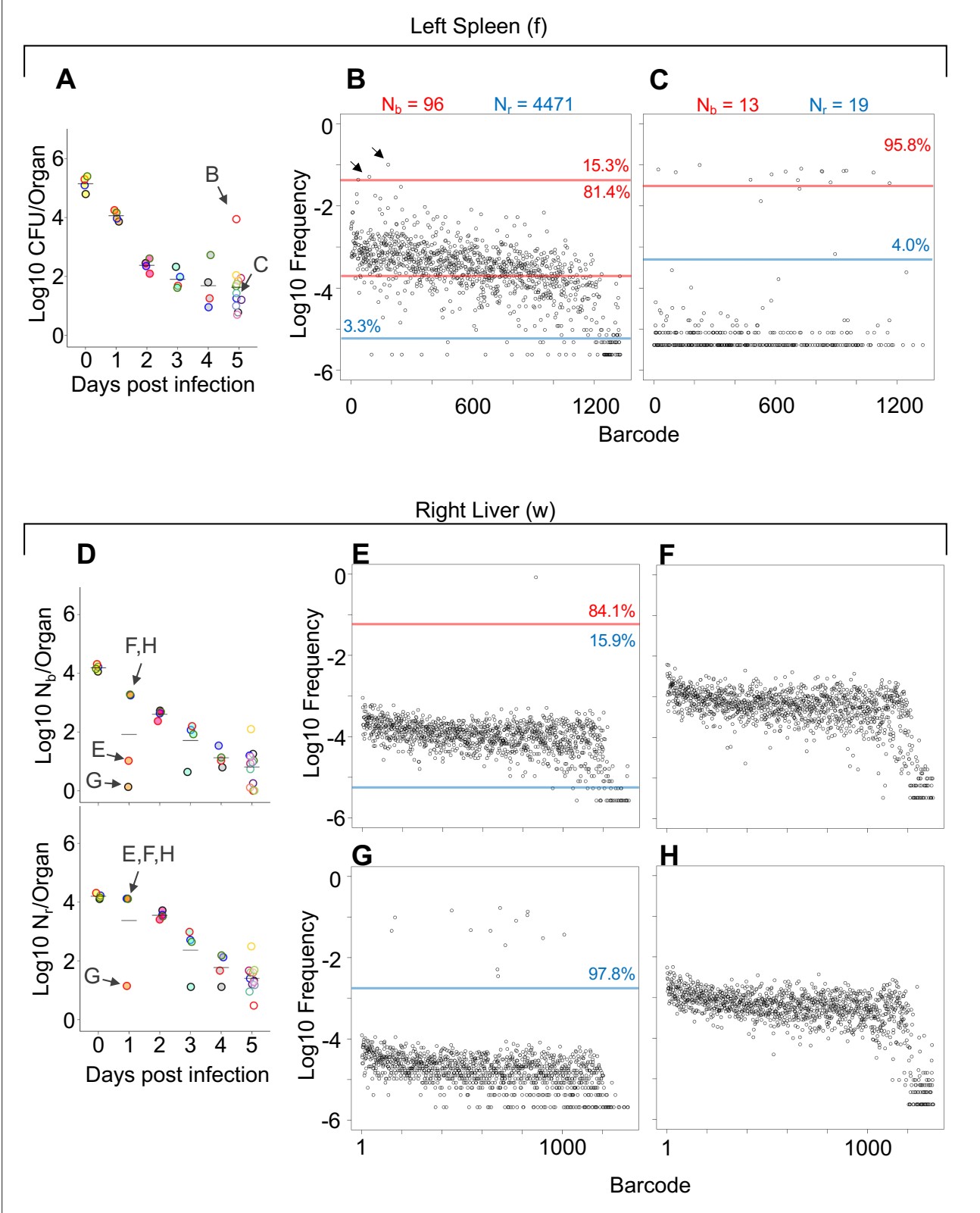

**Figure 2.** Comparisons of barcode distributions in animals with or without clonal expansion events. (**A**) CFU from the left half of the spleen 5 days post infection (dpi); points with the same fill and border color are derived from the same animal. (**B, C**) The underlying barcode distribution found in animals B and C are shown in the respective graphs on the right. The highly increased bacterial burden in animal B relative to animal C can be attributed to both a wider bottleneck and clonal expansion. In the barcode frequency distribution shown in (**B**), a diverse population is evident, as is the clear expansion

*Figure 2 continued on next page*

*Figure 2 continued*

of a few clones (arrows). In contrast, in animal C, $N_b$ and $N_r$ are similar to CFU, suggesting that the population has undergone relatively little replication. (**D**) $N_b$ and $N_r$ plots from the right lobe of the liver 1 dpi. The underlying barcode distributions in animals E, F, G, and H are shown in the respective graphs to the right (in **E–H**). Animals E and G have clonal expansion events; expansion is so substantial in (**G**) (~98% of reads) that the algorithm cannot distinguish the underlying population from noise, and $N_r$ is not substantially higher than $N_b$ for this sample. In contrast, $N_r$ can identify this population in sample E. Red and blue lines denote subpopulations and noise, as in *Figure 1*.

The online version of this article includes the following figure supplement(s) for figure 2:

**Figure supplement 1.** Clonal expansion in the kidney.

within organs are continually being antagonized by host defense processes, consistent with overall decreases in CFU.

In mice with expanded clones in the liver 1 dpi (e.g., *Figure 2D–H*), the dominant barcodes were not detected in other organs or other lobes of the liver, indicating that these expanded clones were locally confined (*Figure 3—figure supplement 2*, asterisks). Similarly, in animals with liver abscesses apparent 4 or 5 dpi, the bacterial populations in the lobe with the abscess were largely distinct from populations in other organs (*Figure 3A*, *Figure 3—figure supplement 6*). Dissection and comparison of individual abscesses from the same animal revealed that these populations were distinct (*Figure 4*), suggesting that abscesses do not originate from the same bacterial population and do not substantially exchange bacteria. Surprisingly, multiple clones, rather than one, comprise a single abscess. In contrast to the liver, bacterial expansion events in other organs were associated with dissemination. The most significant spreading events were observed when ExPEC reached the bile (*Figure 3B*, *Figure 3—figure supplement 5* [mice 17, 20], *Figure 3—figure supplement 6* [mouse 31]). Each of the three animals with CFU in bile harbored a single highly dominant clone (>99% of all reads) and these clones spread systemically to multiple organs; however, the precise organs to which the clone disseminated varied across mice, suggesting that spread of these clones is stochastic (*Figure 3—figure supplements 5–6*). Furthermore, $FRD_{organ-bile}$ was significantly greater than the $FRD_{bile-organ}$, confirming that systemic organs possess relatively more abundant nontransferred populations than bile (*Figure 3C*). More broadly, FRD values across all organs with some relatedness (GD < 0.8) decreased significantly over time, indicating that relatedness at later time points was the result of relatively few barcodes (*Figure 3D*, few barcodes relative to the total barcodes in a sample, not the entire library). Taken together, these results reveal that early genetic relatedness is driven by recent temporal separation (i.e., many barcodes), while later genetic relatedness is typically driven by the dissemination of a few clones.

At higher resolutions, the heuristic value of heatmaps to visualize patterns of dissemination can be illustrated by comparisons of individual mice, such as mice #33 and #34, littermate animals sacrificed 5 dpi (*Figure 3EF*). The GD-based heatmap identifies instances of transfer (bluer colors), and FRD-based heatmaps reveal the fraction of barcodes that contribute to transfer, a metric (in addition to FP) to gauge clonality. Mouse #33, despite having two liver lobes with abscesses containing $10^6$ bacteria, lacked any detectable spread between organs, indicated by the entirely yellow GD heatmap (*Figure 3E*, GD > 0.8). In contrast, mouse 34 had substantial systemic sharing (relatively blue GD heatmap, *Figure 3F*), where a single clone was dominant across all organs (blue/purple FRD heatmap, black colors indicate that no barcodes are shared). Bacterial burdens in the spleen and lungs were also two- to fivefold higher in mouse 34 than mouse 33. The presence of this clone highlights the substantial impact a single bacterium from the inoculum can have on pathogen distribution in the host. Thus, our approach enables detection and quantification of stochastic clonal expansion events; observing such events, which can remain localized or spread, deepens understanding of the origins of within-host pathogen burdens. Collectively, our data reveal that clearance failures and systemic dissemination are mostly attributable to ~0.0001% of the inoculum. The manifestation of these events is highly variable across mice but can fall into three trajectories: (1) complete clearance without dissemination, (2) clonal expansion without dissemination (e.g., liver abscesses), and (3) clonal expansion with dissemination (e.g., bile).

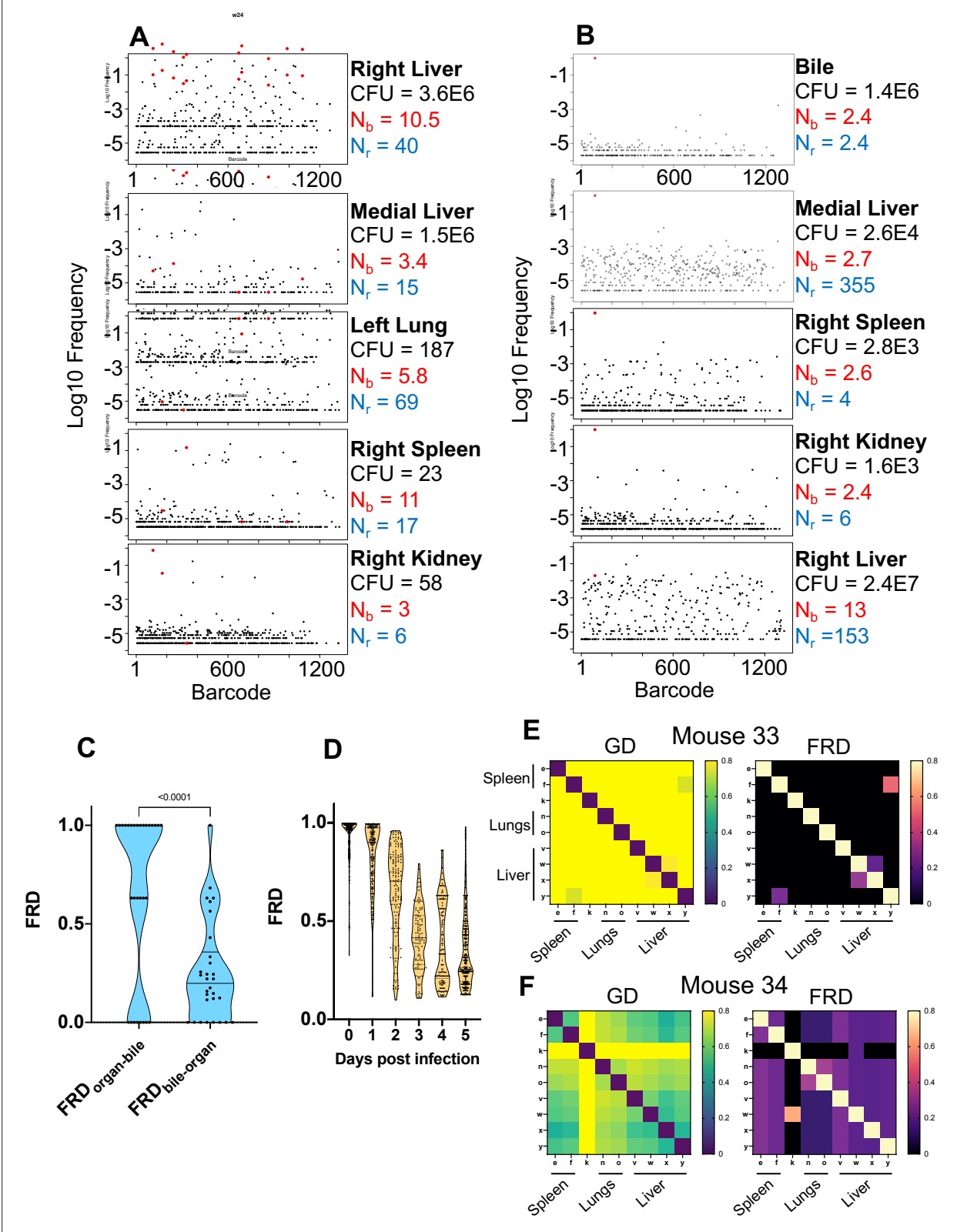

**Figure 3.** Visualizing dissemination patterns. (**A**) Barcode frequency distribution from the right lobe of the liver in a single animal (mouse 24) is shown. This lobe harbored an abscess, which is reflected in the large bacterial burden. The top 10 barcodes are highlighted in red and identified in the four organs below. The relative absence of these dominant barcodes in the other organs reveals that the clones in the abscess did not substantially disseminate to other organs. GD and FRD values resulting from these comparisons are shown in *Figure 3—figure supplement 6*. $N_r$ and $N_b$ values are

*Figure 3 continued on next page*

*Figure 3 continued*

shown adjacent to each sample. (**B**) Same as (**A**) but where the top barcode frequency distribution is from the bile of mouse 20. The most dominant clone is highlighted in red and identified in other organs below. This clone is also dominant in other organs with the exception of the right lobe of the liver, where the clones within an abscess are dominant. (**C**) FRD values resulting from organ comparisons with the bile in all animals with bile CFU (mice 17, 20, and 31). Significantly lower FRD values (Mann–Whitney) when the non-bile organ is set to the reference ($FRD_{bile-organ}$) reveal that the non-bile organ harbors a more substantial nontransferred population, particularly exemplified by liver samples in panel (**B**). (**D**) FRD values for all organs, omitting FRD values of 0 (no detectable relatedness) and 1 (self-organ comparisons). This panel summarizes and quantifies the increasingly purple colors observed across *Figure 3—figure supplements 1–6*. The significant decrease in FRD (Spearman's $r = -0.7$, $p < 0.0001$) indicates that relatedness becomes driven by fewer clones over time. (**E, F**) Heatmaps of GD (left) and FRD (right) are displayed for mouse 33 (**C**) and mouse 34 (**D**). Lower GD values in mouse 34 indicate substantially more sharing of bacteria than in mouse 33. Low FRD values across all organs in mouse 34 reveal that only a few barcodes (i.e., clones) are being shared. FRD = 0 for most organs in mouse 33 because no clones are shared. In (**E, F**), organs displayed are the right spleen (e), left spleen (f), left kidney (k), right lung (n), left lung (o), caudate liver (v), right liver (w), left liver (x), and medial liver (y). Column names in FRD heatmaps represent the organ used as the reference. Heatmaps for all mice and all time points are shown in *Figure 3—figure supplements 1–6*.

The online version of this article includes the following figure supplement(s) for figure 3:

**Figure supplement 1.** Genetic distance (GD) and fractional resilient genetic distance (FRD) 4 hr post infection (hpi) (0 day post infection [dpi]).

**Figure supplement 2.** Genetic distance (GD) and fractional resilient genetic distance (FRD) 1 day post infection (dpi).

**Figure supplement 3.** Genetic distance (GD) and fractional resilient genetic distance (FRD) 2 days post infection (dpi).

**Figure supplement 4.** Genetic distance (GD) and fractional resilient genetic distance (FRD) 3 days post infection (dpi).

**Figure supplement 5.** Genetic distance (GD) and fractional resilient genetic distance (FRD) 4 days post infection (dpi).

**Figure supplement 6.** Genetic distance (GD) and fractional resilient genetic distance (FRD) 5 days post infection (dpi).

## Host bottlenecks are widened by increased pathogen dose and macrophage depletion

The in-depth characterization of ExPEC systemic infection presented above served as a framework that enabled us to begin to investigate the factors and mechanisms that govern these dynamics. Many perturbations in infection models are known to alter bacterial burdens, although quantifying the degree to which changes in CFU result from wider (more permissive) bottlenecks and/or from increased replication has been challenging. The metrics employed in this study can be used to disentangle these processes.

We first assessed how bacterial burden and bottlenecks are influenced by infectious dose, which is fundamentally linked to infection outcome (*Sanchez et al., 2018*; *Schmid-Hempel and Frank, 2007*). The barcoded ExPEC library was i.v. inoculated at two doses (5E6 and 2E7) and the livers and spleens were harvested at 5 dpi. The relatively small difference in the sizes of these inocula further allowed us to assess the sensitivity of CFU and FP to modest differences in dose. Splenic CFU was highly consistent within groups, as observed above, and a fourfold increase in dose resulted in ExPEC burdens ~2.5-fold greater than in the low-dose group (*Figure 5A*). $N_b$ and $N_r$ in the spleen were also relatively similar within the groups, indicating relatively uniform barcode distributions, and both values were significantly larger in the high-dose group, suggesting that there is a general correlation between dose, CFU burdens, and FP sizes in the spleen. Hepatic bacterial burdens were much more variable, consistent with data in *Figure 1—figure supplement 4*. Unexpectedly, there was a reduction in the bacterial burden in the liver at the high dose; a greater incidence of abscesses in the low-dose group (6/10 animals) compared to the high-dose group (2/9 animals) led to a reduction in median CFU at the high dose. Despite the higher hepatic burdens in animals given the low dose, $N_r$ was significantly higher in animals given the high dose, reflecting the role of clonal expansion in elevated CFU. The scaling of FP with dose reveals that a fixed fraction, not number, of bacterial cells from the inoculum survive clearance mechanisms up to 5 dpi. However, in contrast to the spleen, the dramatic replication of very few clones in the liver obscures this relationship from being detected by CFU alone.

To begin to assess the host mechanisms that control bottlenecks, we tested how bacterial burden and bottlenecks are influenced by tissue-resident macrophages, which rapidly sequester bacteria that enter the bloodstream (*Jenne and Kubes, 2013*; *Krenkel and Tacke, 2017*; *Llorente and Schnabl, 2016*). To assess the contribution of these cells in establishing the bottleneck to systemic ExPEC infection, mice were treated with clodronate-containing liposomes, which depletes macrophages from the vasculature, particularly in the liver and spleen (*Van and Sanders, 1994*). The next day, mice were i.v. inoculated with ExPEC at the low (~5E6) dose used above. We chose a dose of clodronate and

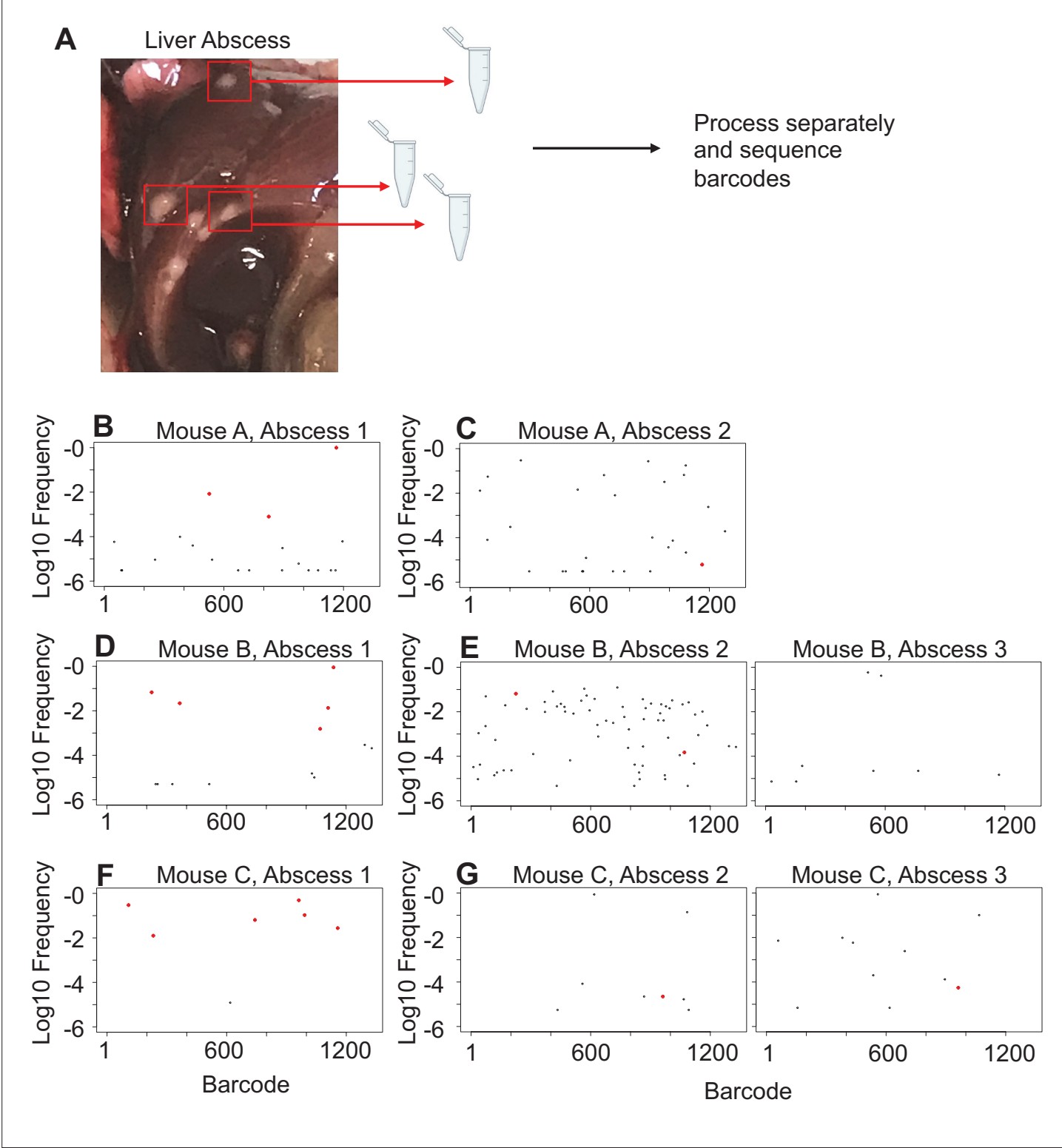

**Figure 4.** Liver abscesses arise from distinct bacterial populations. (**A**) A schematic of the experimental setup is shown. Single abscesses were dissected and separately processed for analysis of barcode frequencies to determine if abscesses in the same animal arose from similar or distinct populations. The panels on the left (**B, D, F**), where dominant barcodes are highlighted in red, correspond to the reference for the graphs on the right (**C, E, G**). These barcodes are then identified on the panels on the right (**C, E, G**). Very few dominant red barcodes in (**C**), (**E**), and (**G**) indicate that populations within individual abscesses are distinct.

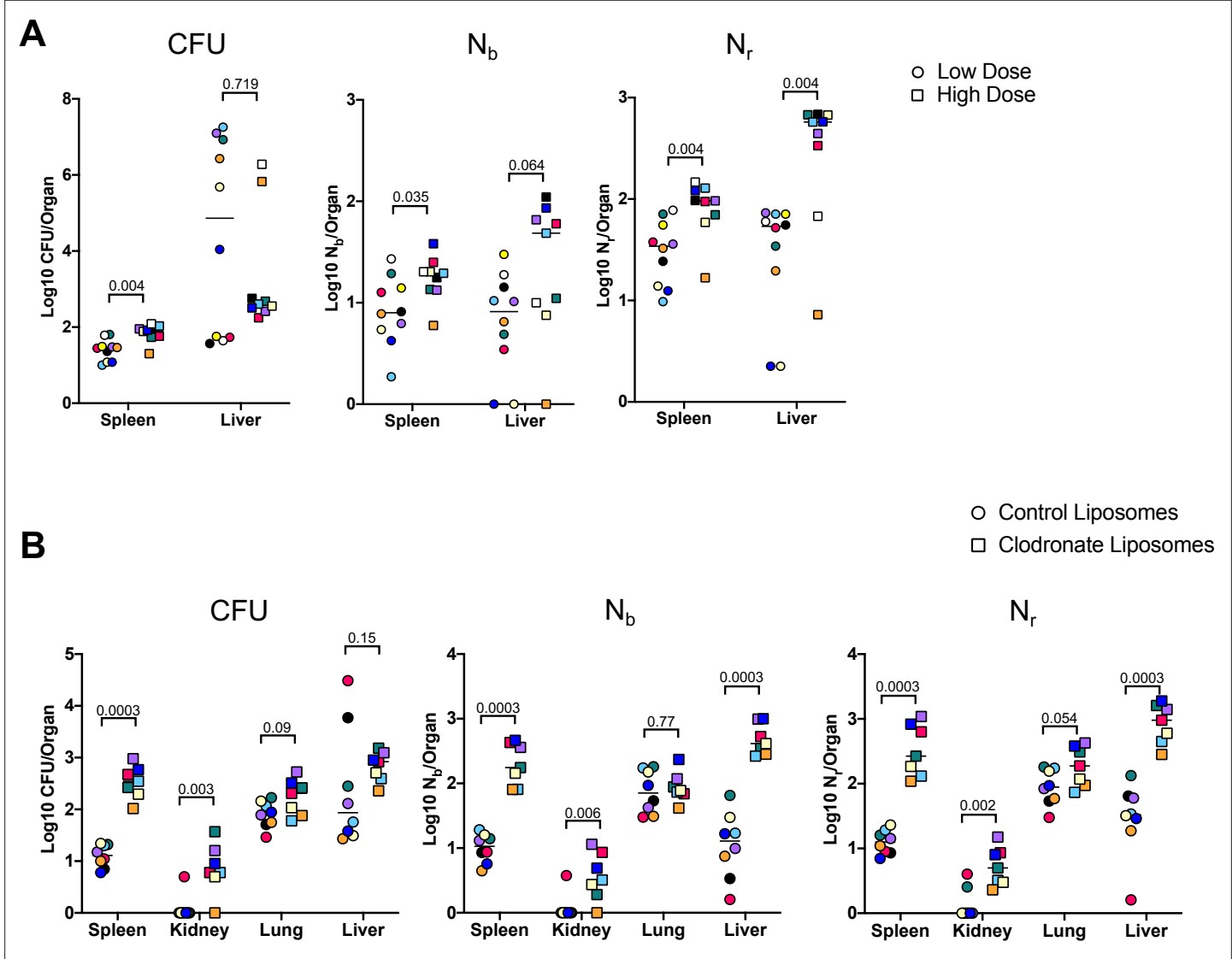

**Figure 5.** Inoculum size and macrophages regulate bottleneck sizes. (**A**) CFU, $N_b$, and $N_r$ values from mice given a low (~5E6, circles) and high (~2E7, squares) dose of extraintestinal pathogenic *Escherichia coli* (ExPEC) intravenously (i.v.) 5 days post infection (dpi). Note that all measurements in the spleen were significantly higher in mice inoculated with a higher dose, but only $N_r$, not CFU, was significantly greater in the liver. Low founding population (FP) values with high CFU indicate that a few clones replicated substantially. Points with the same color and shape are derived from the same animal. (**B**) CFU, $N_b$, and $N_r$ values from mice treated with control liposomes (circles) or clodronate-containing liposomes (squares). CFU is significantly increased in the spleen and kidneys, but not the lung and liver. FP is significantly greater after clodronate treatment in the liver. Points with the same shape and color are from the same animal. The two animals with high CFU in the liver result from the expansion of very few clones (high CFU, low FP). Statistical significance for each comparison represents q-values from multiple Mann–Whitney tests.

control liposomes that only partially depletes macrophages (*Tavares et al., 2017*) because at higher, commonly used clodronate doses, animals rapidly succumb to infection (*Pollitt et al., 2018*). There was an increase in splenic CFU 5 dpi in animals treated with clodronate, whereas this difference was more subtle in the liver, lungs, and kidneys (*Figure 5B*). $N_b$ and $N_r$ calculations revealed that macrophage depletion increased FP sizes in the liver, spleen, and kidneys, although the difference in the lung was not statistically significant. The significant increase in FP, but not CFU, after clodronate treatment in the liver is explained by two animals in the control group with the dramatic expansion of very few clones (*Figure 5B*, red and black circles). Since clodronate treatment increased both FP and CFU, macrophage depletion widens the bottlenecks in the liver, spleen, and kidney. Taken together with the dosing experiment, these observations underscore how stochastic expansion of clones can obscure detection of biologically significant signals, highlighting the utility of using barcoded libraries during

experimental infection for even subtle changes in experimental parameters. In addition to being able to detect these signals, barcoding approaches can reveal whether elevation in CFU burdens is attributed to heightened bacterial replication within organs or wider host bottlenecks. In this model, increased dose and macrophage depletion widened bottlenecks in the host but did not increase the levels of bacterial replication.

## Bacterial genetic analyses disentangle organ-specific and temporal infection outcomes

Our STAMPR-based analyses uncovered several patterns of pathogen population trajectories in infected hosts, including eradication early during infection, survival/persistence until late in infection, and clonal expansion with or without dissemination. Moreover, each of these trajectories had organ-specific patterns. Since bacteria within these different trajectories likely encounter distinct host niches, we set out to identify bacterial genetic factors that distinguish these environments. We further reasoned that identification of organ-specific bacterial genes required for early survival, but not for subsequent expansion or persistence, would (1) signify that early and late survival/expansion are at least in part separable and (2) identify the pathways that distinguish them. To identify early pathogen survival determinants, we analyzed an ExPEC transposon library composed of ~150,000 unique transposon mutants 1 day after i.v. inoculation. As expected (*Figure 1—figure supplements 2–4*), we detected highly dominant mutants (i.e., clones) in the livers and spleens of 3/5 mice (*Figure 6—figure supplement 1*). Since these three animals also had markedly high bile CFU, these clones are likely also present in the bile.

Comparisons of the abundances of gene insertion frequencies in samples isolated from the liver and an input that was replated on LB were carried out to identify processes that contribute to early ExPEC survival using a new analytic pipeline (see Materials and methods) that simulates infection bottlenecks. There was a predominance of cell envelope homeostasis genes and gene clusters that were underrepresented in liver samples, including *mlaFEDC*B (*Malinverni and Silhavy, 2009*; *Ekiert et al., 2017*), ftsEX (*Du et al., 2016*; *Yang et al., 2011*), envZ/ompR (*Hall and Silhavy, 1981*), and many LPS biosynthetic genes (*Figure 6A*), suggesting that cell envelope homeostasis across multiple pathways is required for extraintestinal survival. Another coherent set of underrepresented genes in the liver included *manA, nagAB,* and *pgi*, which convert mannose-6-phosphate, n-acetylglucosamine-6-phosphate (GlcNAc-6P), and glucose-6-phosphate into fructose-6-phophate, respectively (*Paixão et al., 2015*; *Figure 6B*). GlcNAC-6P, the substrate of NagA, is generated in *E. coli* through two mechanisms (*Uehara and Park, 2004*). In the first, extracellular GlcNAc is phosphorylated and imported through the mannose phosphotransferase system. In the second, GlcNAc is released as a peptidoglycan degradation product, and is phosphorylated by NagK (GlcNAc kinase). Interestingly, *nagK* was not underrepresented in the liver, suggesting that the host is the source of GlcNAc. Taken together, the screen revealed that several cell envelope homeostasis pathways are important for early survival in the liver and an additional requirement for fructose-6-phosphate during systemic infection.

To investigate how the genetic requirements for ExPEC survival vary during infection, we chose a subset of the most underrepresented genes/pathways for early hepatic survival to test for survival across organs and over time. Traditionally, competition between WT and mutant strains is performed. These studies rely on CFU plating to determine the relative abundance of the mutant strain before and after infection (*Karlinsey et al., 2019*; *Wang et al., 2014*; *Warr et al., 2019*; *Subashchandrabose et al., 2013*). However, as we showed above, this approach would likely be confounded by early stochastic clonal expansion events (particularly in the liver). Instead, we adopted a barcoding strategy similar to previous studies (*Warr et al., 2019*; *Hubbard et al., 2016*). In-frame deletions of *mlaC* (RS18735), *ftsX* (RS20130), *yejM* (RS12925), *manA* (RS09455), *nagA* (RS03530), and *nagK* (RS06530) were created and each strain, along with the WT, was barcoded with three tags and pooled, resulting in an inoculum comprising a mixture of 21 tags. The Δ*nagK* strain served as a 'negative control' as it was not underrepresented in our screen (*Supplementary file 1*). This library was i.v. inoculated and the bile, spleen, right kidney, lungs, and all four lobes of the liver (separately) were harvested 1 and 5 dpi. Barcodes in each sample were then sequenced to determine the abundance of all 21 tags, and therefore each mutant.

1 dpi in the liver, stochastic clonal expansion was observed and manifest as a dramatic increase in the abundance of a few barcodes and the corresponding marked decrease in the abundance of

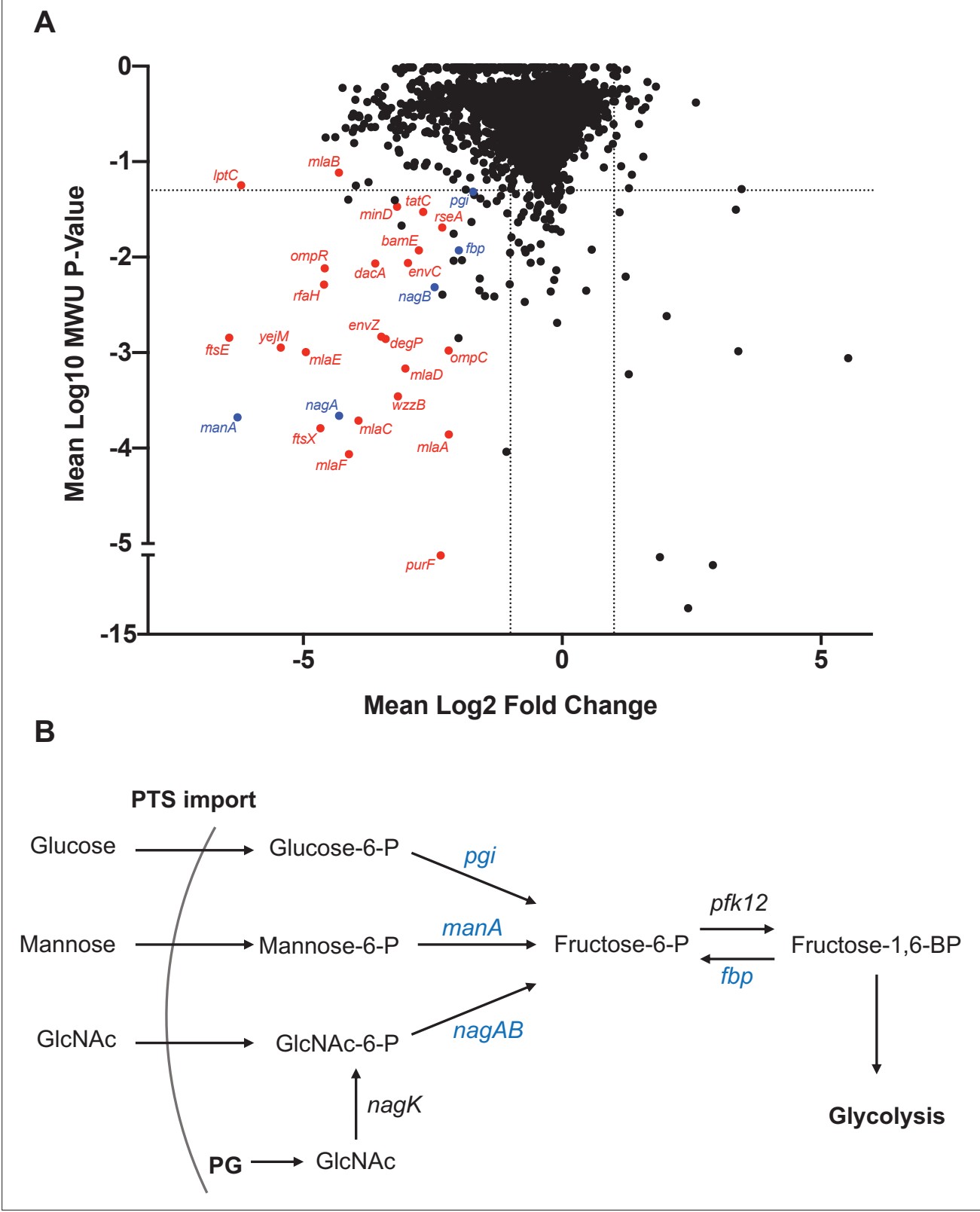

**Figure 6.** Genes that promote extraintestinal pathogenic *Escherichia coli* (ExPEC) survival in the liver. (**A**) Inverse volcano plot of fitness of ExPEC genes in the liver. Genes shown in red are involved in cell envelope homeostasis, and genes shown in blue are involved in generating fructose-6-phosphate, and their roles are schematized in (**B**). (**B**) Pathways to convert sugars to fructose-6-phosphate. GlcNAc generated from peptidoglycan (PG) degradation is converted to GlcNAc-6P by *nagK,* which is not among the underrepresented genes in (**A**).

*Figure 6 continued on next page*

*Figure 6 continued*

The online version of this article includes the following figure supplement(s) for figure 6:

**Figure supplement 1.** CFU and population structure from in vivo transposon mutant screen.

all other tags in the sample, even those from the same genotype. Thus, clonal expansion explains the large variance in fitness measurements. For example, one animal had a substantial expansion of one of the three tags barcoding the *mlaC, manA,* and *nagA* mutants in the caudate lobe of the liver (*Figure 7A*, red asterisks), within an associated drop in the abundance of all other barcodes from the animal, even the other barcodes marking the same mutants. In the liver, all genes found to be underrepresented in the transposon screen displayed fitness defects. However, the magnitude of the defects in the *mlaC, ftsX, yejM,* and *nagA* mutants were far more pronounced than that observed in the *manA* mutant, which possessed modest but consistent defects relative to the WT. Clonal expansion events likely prevented *manA* from reaching statistical significance (*Figure 7A*). Similar patterns of mutant fitness defects were observed in the spleen and the kidneys, although *manA* was not statistically different than the WT 1 dpi in the kidneys (*Figure 7—figure supplement 1*). Intriguingly, there was no difference between Δ*manA* and Δ*nagA* compared to Δ*nagK* and WT in the lungs, indicating that these genes have organ-specific importance for ExPEC survival and reveals that the lung microenvironment has particularly distinct selective effects (*Figure 7C*). The Δ*nagK* negative control was identical to the WT across all organs 1 dpi. The differential importance of *nagA* and *nagK* confirms our transposon insertion sequencing (TIS) results and provides further evidence that the source of GlcNAc-6P is the host.

We then assessed the role of these genes for survival at 5 dpi, during the persistence and expansion phase. In this cohort, 5/8 animals had at least one lobe of the liver with an abscess, and therefore several orders of magnitude more bacteria than lobes without abscesses (*Figure 7—figure supplement 1*). Since each liver lobe was segregated, however, we grouped lobes with and without abscesses separately for analysis; this grouping enables assessment of genetic requirements for persistence (no abscess) or for expansion (abscess). In addition, we omitted the kidneys from this analysis as too few bacteria were recovered 5 dpi (*Figure 7—figure supplement 1*).

The Δ*mlaC*, Δ*ftsX,* and Δ*yejM* strains were significantly depleted in the spleen, liver (with or without abscesses), and lungs relative to the WT at 5 dpi, further highlighting the essentiality of the cell envelope-related process to survival within these organs (*Figure 7B–D*). In contrast, despite having clear defects at 1 dpi, Δ*nagA* and Δ*manA* were indistinguishable from WT and Δ*nagK* in the spleen and the liver at 5 dpi, regardless of the presence of abscesses. An intriguing comparison is Δ*nagA* and Δ*mlaC,* which display nearly identical fitness defects in the liver 1 dpi but clearly differ by 5 dpi (with or without abscesses). To our surprise, Δ*nagA* was significantly more abundant in the lungs than the WT or Δ*nagK* at 5 dpi (*Figure 7C*). Therefore, cell envelope and sugar metabolism genes likely contribute to distinct mechanisms that facilitate ExPEC survival, in the face of host factors that fluctuate over time and between organs. These results further reveal that the pathogen factors that are required for the initial establishment of infection differ from those required to survive later or clonally expand. Specifically, ExPEC requires hexose metabolism to establish populations within specific organs, but this pathway is not required by the subpopulations that ultimately persist later in the infection or clonally expand, suggesting that establishment, persistence, and expansion depend on distinct pathogen pathways. These data further highlight how understanding within host infection dynamics provides critical contextualization of pathogen virulence or fitness factors that vary in their importance across time and space.

## Discussion

Here, using pathogen barcoding and a high-resolution approach for comparing barcode frequency distributions, we describe the complex infection dynamics of systemic ExPEC infection (summarized in *Figure 8*). Although we found that ExPEC is largely cleared by most organs, we observed organ-specific population dynamics and discovered that clearance failures are pervasive, particularly in the liver. These failures manifest as dramatic expansions of very few bacterial cells from the inoculum. Some expansions are not easily revealed by CFU alone (e.g., clonal expansion at 1 dpi in the liver), but our analytic approach readily detects and quantifies these events in addition to contextualizing

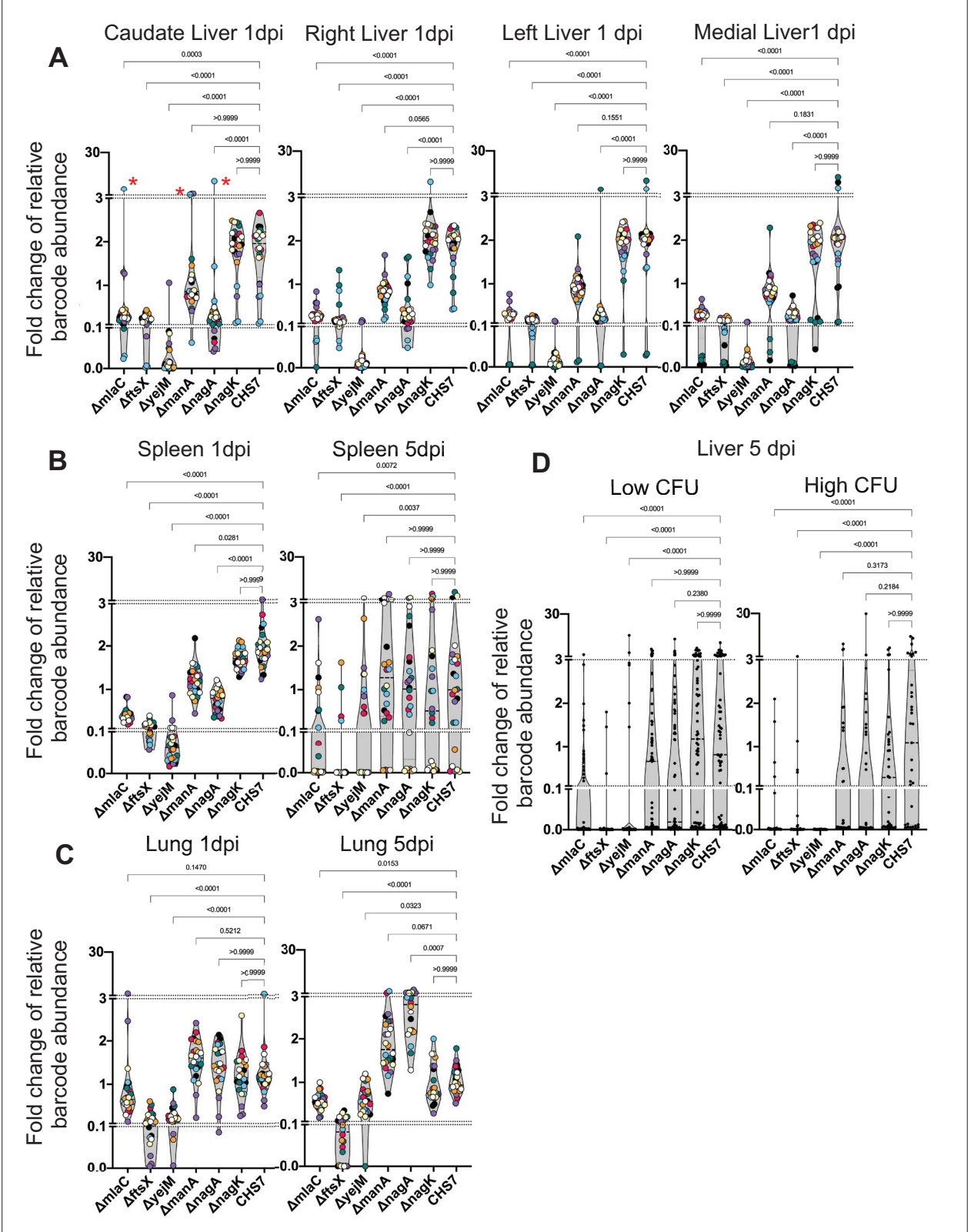

**Figure 7.** Distinct genetic requirements for extraintestinal pathogenic *Escherichia coli* (ExPEC) survival/growth across organs and time. (**A**) Changes in barcode frequencies for WT and six mutants after coinfection are shown for liver 1 day post infection (dpi). Red asterisks represent an example of clonal expansion. (**B, C**) same as (**A**) for spleen (**B**) and lungs (**C**) 1 and 5 dpi. (**D**) Same as (**A–C**) but for liver samples 5 dpi, separated into lobes with or without abscesses. *Figure 7—figure supplement 1* displays this data separated by lobe, in addition to data from the kidneys. Within each graph, points with

*Figure 7 continued on next page*

*Figure 7 continued*

the same color were derived from the same animal. Each strain is represented by three barcodes, so within each graph, each color is represented 21 times. Dotted lines depict breaks in the y-axis that are useful to visualize the influence of clonal expansion. p-Values (Dunn's multiple comparisons test) are shown relative to the WT.

The online version of this article includes the following figure supplement(s) for figure 7:

**Figure supplement 1.** CFU and additional organs from *Figure 7*.

them with populations at other sites. During experimental perturbations, our analyses also readily distinguished when increases in bacterial burden are due to widening of bottlenecks or to increased bacterial replication. We found that, despite clonal expansion, host bottlenecks were highly consistent between animals across all organs. However, bottlenecks are highly sensitive to subtle organ-specific changes in experimental parameters, including inoculum size and macrophage depletion. Furthermore, we identify a potential role for bacterial sugar metabolism that distinguishes early extraintestinal survival with later persistence or clonal expansion. Our analysis of population dynamics can also

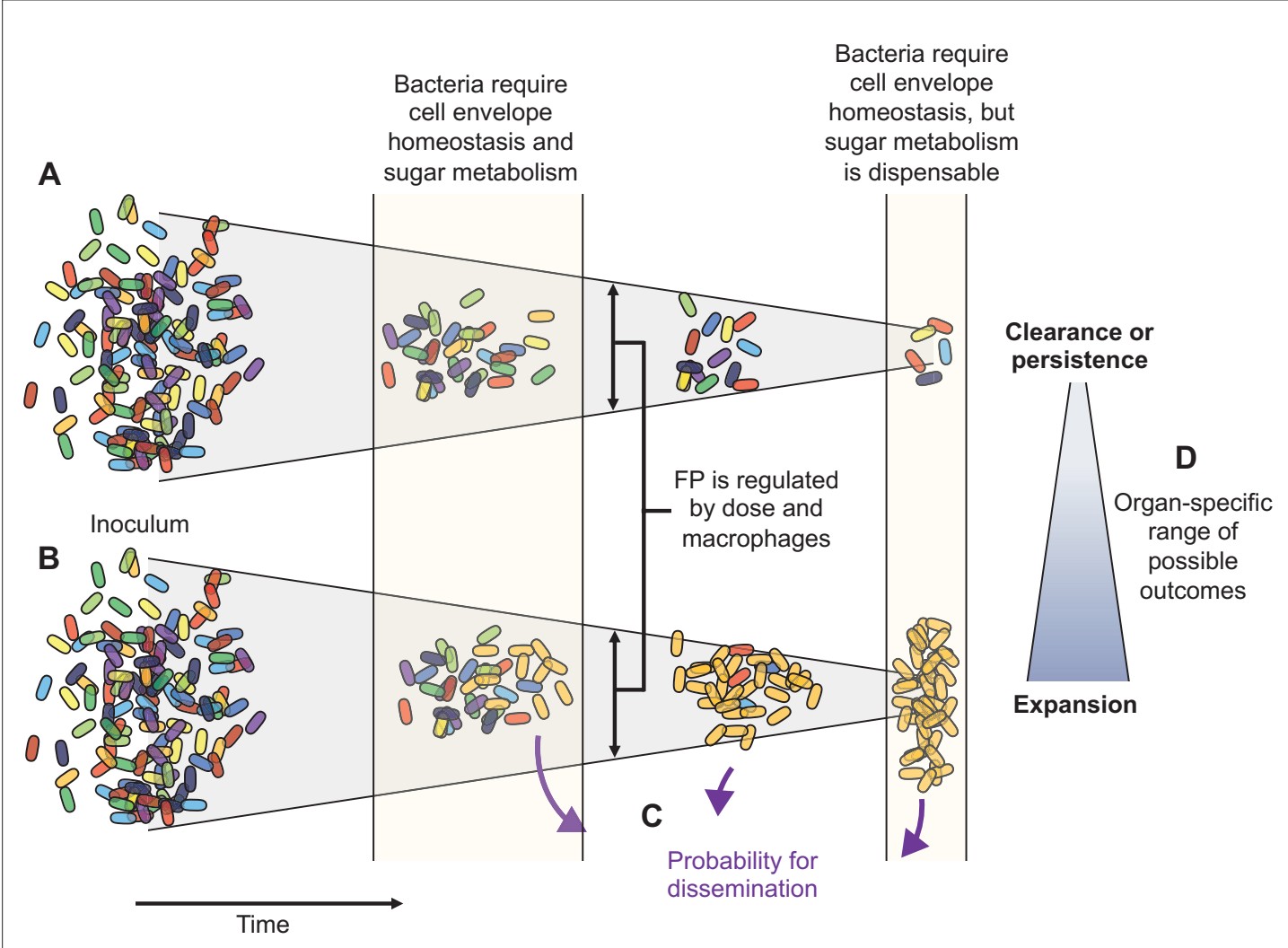

**Figure 8.** Schematic summary of key findings. (**A**) and (**B**) schematize the most distinct trajectories of extraintestinal pathogenic *Escherichia coli* (ExPEC) population dynamics identified – one where bacteria are nearly cleared (**A**) and one where they expand clonally (**B**). Additional intermediate categories are also observed when clonally expanded populations disseminate systemically and alter bacterial burdens and population structure in distal organs (**C**). The probability of dissemination depends on the organ where clonal expansion occurred (**D**); for example, there was a much larger likelihood of disseminating when there was clonal expansion evident in bile versus from the liver. Every animal displays a unique pattern of expansion and dissemination.

refine transposon-based studies of genetic requirements by contextualizing bacterial genes required for infection in the setting of temporal, organ-specific, and stochastic infection outcomes. In addition, understanding the population dynamics of an infection model facilitates the optimization of experimental designs in infection studies.

Infection bottlenecks are critical determinants of infection outcomes. This notion was appreciated as far back as the 1950s with the 'independent action' hypothesis, which posits that each organism from the inoculum possesses equal probability to cause infection (**Moxon and Murphy, 1978**; **Rubin, 1987**; **Shechmeister, 1950**). The idea was based on observations that at lower infectious doses, but above the minimum infectious dose, the bacterial population in the animal was more likely to be derived from a single organism, whereas at higher doses the population was more likely to be derived from a several organisms from the inoculum. Through the lens of bottlenecks, this hypothesis can be reinterpreted to imply that each bacterium has an equal probability of passing through the bottleneck. If the nature of the bottleneck is fractional (i.e., a fraction the inoculum passes through), increasing the dose increases the number of organisms that go on to initiate infection. Our finding that FP, but not CFU, scales with infectious doses is consistent with the independent action hypothesis, suggesting that bottlenecks, rather than replication, are more directly influenced by the infectious dose. The current study sets the stage for future investigations to further explore the relationships between dose, FP, replication, and overall infection outcomes in diverse pathogens.

Our findings highlight the differential capacities of host organs to clear bacteria. In general, most organs continuously restricted the pathogen population over time, indicated by decreasing N, values over time. A notable exception to this trend was the lungs, where FP values largely plateaued by 2 dpi, explaining why this organ had among the highest FP sizes 5 dpi. In addition, clodronate had no effect on CFU in the lung and some mutants defective for survival in the liver were more abundant than WT in the lung. These three lines of evidence indicate that the lung microenvironment is less hostile to ExPEC, perhaps reflecting distinct cellular niches (e.g., different immune cell compositions and nutrient availabilities) occupied by ExPEC in the lungs relative to the liver or spleen. Furthermore, we discovered that ExPEC forms visible abscesses in liver at much higher frequencies than expansion events in other organs. Thus, the liver and lungs appear to differ markedly in their clearance defects; the liver appears more permissive for ExPEC replication, whereas the lung appears more permissive for bacterial survival. More broadly, STAMPR enables detection of these two distinct processes.

*E. coli,* and more recently *Klebsiella pneumoniae,* are among the leading causes of liver abscesses in humans, and patients with liver abscesses with associated bacteremia are at higher risk for mortality (**Meddings et al., 2010**). The infection model described here may therefore be a tractable experimental system to dissect the host and pathogen factors that underlie *E. coli* liver abscess formation. Since there are very few detectable bacteria in blood, i.v. administration of ExPEC may not represent a good model of human bloodstream infection per se; instead, this model may be more relevant for investigating the efficiency of organ clearance and capture of bloodborne bacteria in addition to within-organ replication. In humans, the source of bacteria in blood varies widely. Although tail vein injection in mice models a very specific route, the fate of bacteria after reaching the bloodstream regardless of the initial entry point may be similar. Furthermore, the i.v. doses used here do not directly model doses likely encountered by humans; however, mice are substantially more resistant to LPS than humans, and higher doses in mice may mimic the lower doses humans are likely to encounter (**Copeland et al., 2005**). The limitations and advantages of various bloodstream infection models have been recently reviewed (**Holmes et al., 2021**).

Very few studies to date have reported on the population dynamics of systemic infections. In one study, i.v. inoculation of a *S. aureus* pool consisting of three differentially marked strains revealed several trends observed here (**Pollitt et al., 2018**). Although the use of three markers limited the resolution, this study of *S. aureus* dissemination also found that bottlenecks (measured as the proportion of organs with less than two strains) widened in size with infectious dose and macrophage depletion. However, unlike ExPEC where expansion of clones in liver abscesses remained locally confined, *S. aureus* dissemination from the liver was thought to result in clonal abscesses in the kidney. In general, abscess formation is much more consistently observed for *S. aureus* (**Sx et al., 2014**; **Cheng et al., 2009**), whereas this outcome is apparently stochastic in this ExPEC model. Thus, pathogen-specific factors exert distinct roles in the determination of the organ-specific outcomes of clonal expansion events. Another study examining systemic *P. aeruginosa* population dynamics with STAMP found that

the pathogen consistently replicated in the bile and these cells became the source of bacteria in the intestinal tract (*Bachta et al., 2020*). In ExPEC, the most dramatic instances of dissemination also occurred in animals with ExPEC in bile. The FP of the bile population in our study was ~1, whereas we calculated that the *P. aeruginosa* FP was ~10 times greater (*Hullahalli et al., 2021*). The more restrictive bottleneck for ExPEC potentially explains why transit to the bile is stochastic with this pathogen. The mechanisms by which pathogens such as ExPEC and *P. aeruginosa* spread systemically from bile require further investigation but may be linked to the event(s) by which the pathogen seeds the bile to begin with.

The bacterial genetic determinants of systemic infection are of great interest as these may inform development of therapeutics for the treatment of these severe infections (*Armbruster et al., 2019*; *Subashchandrabose et al., 2013*; *Subashchandrabose et al., 2016*; *Anderson et al., 2018*; *Anderson et al., 2017*). In models of overwhelming bacteremia, these genetic determinants often encode surface structures, such as the capsule in K1 ExPEC (*McCarthy et al., 2018*). Similarly, we found that most of the underrepresented genes in the liver and spleen encoded components of the cell envelope. Unexpectedly, there was little overlap between genes we identified to be underrepresented in the liver or spleen and genes identified as underrepresented in a previous study using a similar experimental system (*Subashchandrabose et al., 2013*). A potential explanation for this discrepancy is that Subashchandrabose et al. performed their experiments in a different mouse background (CBA/J), and different strains of mice impose distinct barriers to infection, altering the pathogen genetic requirements for infection (*Cunrath and Bumann, 2019*). In addition, it is unknown whether clonal expansion occurs in CBA/J mice. Our findings suggest that understanding population dynamics prior to determining genetic requirements can refine identification of genetic dependencies in the presence of genotype-independent expansion events. More broadly, the use of isogenic libraries (STAMPR) can complement mutant libraries (TIS) by providing important information about bottlenecks and can contextualize genetic requirements in the landscape of population dynamics. Furthermore, mutant libraries complement isogenic libraries by revealing further distinctions (e.g., bacterial genetic determinants) that distinguish broader population-level phenomena in the host, such as replication and persistence.

Analysis of barcode frequency distributions permits a more quantitative and deeper dissection of population dynamics in experimental infection models beyond enumerating bacterial burdens within organs. The eight metrics we used (*Figure 1A*, *Hullahalli et al., 2021*) to compare pairs of organs within an animal allow for delineation of patterns of expansion and dissemination. For example, we show how some expanded clones (i.e., in liver abscesses) appear to have relatively little influence on barcode frequencies in other organs in the animal (*Figure 3A*). In stark contrast, *Figure 3B* reveals an example in which a single clone in the bile profoundly alters the pathogen distribution in multiple organs. In many cases, we observed that CFU values were similar to FP values (*Figure 1I*, *Figure 1— figure supplements 2–4*), suggesting that the respective bacterial population had not undergone substantial net replication. Heatmaps enable direct visualization of dissemination patterns and the degree to which these dissemination events manifest within an organ. Heatmaps from animals 5 dpi revealed marked heterogeneity in dissemination patterns that vary across organs (*Figure 3—figure supplement 6*). Furthermore, these metrics reveal the variability between animals both in terms of clonal expansion within organs, bottlenecks between organs (*Figure 1—figure supplements 2–4*), and dissemination within and between organs (*Figure 3—figure supplement 6*). Beyond comparisons between animals, our metrics can be applied to single animals and revealed that stochastic expansion and dissemination of very few cells from the inoculum was pervasive.

Early stochastic clonal expansion events were detected in every experiment in our study using three independent libraries and distinct approaches (STAMPR, transposon libraries, and barcoded mutants). These expansion events underlie variation in infection outcomes. While dissection of the mechanism(s) that account for this stochasticity is beyond the scope of this study, our observations suggest that several factors account for and modulate clonal expansion. Such events appear to fall into two functional categories – those that are due to dissemination (e.g., from bile) and those that are not (e.g., liver abscesses). In the latter, we observed an apparent inverse correlation with infectious dose, although more animals are required to reach statistical significance (*Figure 5A*). This trend suggests that more robust early immune induction (presumably elicited with a higher dose) disproportionately clears bacteria, a hypothesis that has also been generated by mathematical modeling

(*Ellner et al., 2021*; *Duneau et al., 2017*). Additionally, we observed apparent lobe (more frequent in the right lobe) and topological (all on the surface) biases in liver abscess formation, raising the possibility that anatomic differences between lobes and their relative locations contribute to apparently stochastic outcomes as some lobes may be more likely to be seeded (due to blood flow) and more permissive to bacterial replication. Furthermore, clonal expansion explains why bacterial burden is highly variable in ExPEC and may also explain high variability in systemic infection outcomes in other bacteria (*Crépin et al., 2018*).

Several lines of evidence suggest that causative genotypic changes are unlikely to be linked to pathogen clonal expansion in liver abscesses, but we cannot definitively exclude this possibility. Expanded clones in the liver arise as early as 1 dpi, which is relatively early in the infection processes when relatively little bacterial replication has occurred. If causative mutants existed in the inoculum prior to infection (e.g., in the colony that was picked to generate the barcoded strain), we would expect the same set of barcodes to expand; however, different sets of barcodes expanded across all three experimental barcoded libraries (e.g., abscess clones in *Figure 4* are different). If we assume a fixed mutation rate, then we would expect a similar number of clones to expand across animals, as observed in *Jasinska et al., 2020* and in *Vasquez et al., 2020*; however, we observe a bimodal distribution of either 0 expanded clones or ~5–100 expanded clones. Furthermore, since liver abscesses are relatively small and discrete but consist of multiple clones, causative mutations would have had to arise in different bacterial cells that were also clustered together in physical space, which seems extremely unlikely. Notably, despite tight bottlenecks, we predict that beneficial mutations would accumulate if bacteria isolated from abscesses were used to inoculate another set of mice and the process repeated for several passages, as has been observed with *S. aureus* passages in sheep over hundreds of days (*Bacigalupe et al., 2019*).

The STAMPR analytic framework (*Hullahalli et al., 2021*) applied here to study infection population dynamics revealed how very few cells in a large inoculum disproportionally contribute to pathogen burdens in different host organs. By monitoring patterns of dissemination, we uncovered that these few cells can drastically alter pathogen distributions in distal organs. Our findings suggest that perturbations that impede the expansion of the few cells that clonally expand could profoundly alter the course of infection. Finally, our observations of organ-specific pathogen population dynamics and genetic requirements suggest that investigation of host tissue-specific factors that regulate these dynamics, perhaps centered around bacterial metabolism of host-derived sugars, may uncover novel facets of host-pathogen interactions.

## Materials and methods

### Media and antibiotics

Bacteria were routinely cultured in LB at 37°C. Additional compounds and antibiotics were used at the following concentrations: kanamycin 50 µg/ml, chloramphenicol 20 µg/ml, carbenicillin 50 µg/ml, and diaminopimelic acid (DAP) 300 µM. Bacteria scraped from plates were resuspended in PBSG (PBS 25% glycerol).

### Animal experiments

All animals used in this study were 8–10-week-old female C57BL/6J mice obtained from Jackson Laboratories and housed for 3 days prior to handling. Defined volumes of frozen cultures were thawed and resuspended in PBS for i.v. inoculations. Prior to inoculation, mice were gently warmed on a heating pad and restrained in a Broome style restrainer (Plas Labs), and 100 µl of bacteria was injected into the lateral tail vein with a 27G needle. Cages were then left undisturbed until the time of sacrifice. For clodronate or control liposome (Encapsula) treatment, 50 µl was injected i.v. 1 day prior to infection. We did not directly quantify the number of macrophages in this study. For all experiments in the study, the bile in the gallbladder was aspirated with a 30G needle into 100 µl of PBS, after which the gallbladder was surgically removed prior to harvesting the liver. We detected bile CFU in four animals across all experiments in this study. Three are detailed in the first experiment (*Figure 1— figure supplement 2*), and the fourth was present in the clodronate-treated cohort but was moribund prior to the time of sacrifice and was therefore excluded from analysis. The 'left lung' samples described in the first experiment (*Figure 1—figure supplements 2–4*) includes the post-caval lobe.

All 'blood' samples are derived from a 200 µl cardiac bleed. After sacrifice, organs were homogenized with two 3.2 mm stainless steel beads (Biospec) for 2 min in PBS and plated on LB+ kanamycin plates.

## Bacterial strain construction

CFT073 *rpoS*⁻ (**Welch et al., 2002**; **Hryckowian and Welch, 2013**; **Hryckowian et al., 2015**) and CFT073 *rpoS*⁻ Δ*mlaC* were conjugated (500 µl donor with 500 µl recipient overnight on a 0.45 µM filter) with *E. coli* donor strain MFD λ pir (**Ferrières et al., 2010**) harboring a pDS132 (**Philippe et al., 2004**) derivative containing appropriate homologous recombination fragments to facilitate repair of the *rpoS* allele. Transconjugants were selected on LB+ chloramphenicol, re-streaked on LB+ chloramphenicol, and cultured overnight with chloramphenicol. 300 µl was diluted in 3 ml of LB with 10% sucrose, cultured overnight at 37°C, and plated for single colonies on LB with 10% sucrose. Sanger sequencing was used to identify colonies with the repaired *rpoS* allele. Positive colonies were further confirmed by whole-genome sequencing through MiGS (Pittsburgh, PA). The CFT073 strain with the corrected *rpoS* allele is designated as CHS7.

The same sucrose-based allele exchange method was used to create in-frame deletions of *manA*, *yejM* (C-terminal domain only), and *ftsX* from CHS7. For deletion of *nagA* and *nagK*, we adopted a CRISPR-based counterselection approach used in **Jiang et al., 2015** with several modifications. We first introduced *bla* and *sacB* from pCVD442 (**Donnenberg and Kaper, 1991**) into pSU-araC-Cas9, creating plasmid pCAS, which contains an arabinose-inducible *cas9*. A separate plasmid containing the guide RNA was generated from pTargetF (**Jiang et al., 2015**) by introducing homology fragments in addition to *sacB, cat,* and conjugation machinery from pDS132, creating a family of plasmids referred to as pGuide. Derivatives of pGuide were created with the appropriate spacer targeting the gene of interest and homology fragments. To generate deletions in a target gene, the recipient strain was first transformed with pCAS, which harbors a beta-lactamase gene conferring carbenicillin resistance. pGuide was then introduced into a pCAS-carrying recipient strain by conjugation and transconjugants were selected on plates containing chloramphenicol, carbenicillin, and 0.2% arabinose overnight at 37°C. The transconjugants were then scraped and replated on plates containing chloramphenicol, carbenicillin, and 0.2% arabinose overnight at 37°C. Single colonies from this second passage were streaked again on plates containing chloramphenicol, carbenicillin, and 0.2% arabinose. Single colonies from this passage were then streaked on LB plates containing 10% sucrose and lacking NaCl to counter-select both plasmids, which was confirmed by loss of the resistance markers. This protocol was initially optimized by targeting *lacZ*, where blue/white screening could be used to determine the steps at which successful edits have been made. Successful deletions were confirmed by PCR and, after a subsequent colony purification, a second PCR. In addition, both the *nagA* and *manA* mutants were unable to grow in minimal media supplemented with GlcNAc or mannose as the sole carbon source, respectively.

## Construction of barcoded ExPEC library

The plasmid used to introduce barcodes into CHS7 was created by amplification of a kanamycin fragment adjacent to *lacZ* with a primer containing 20 random nucleotides (Integrated DNA Technologies). This fragment was cloned into pDS132 using the NEB HiFI DNA Master Mix (New England Biolabs) and transformed into MFD λ pir and plated on LB+ kanamycin + DAP, yielding pDS132-STAMPR. The cloning reaction was scaled up to yield 70,000 colonies. Ten colonies were individually Sanger-sequenced to confirm the presence of a single, unique barcode. Cells were then frozen into individual aliquots.

To introduce barcodes into CHS7, 30 µl aliquots of the barcode donor were thawed in 3 ml LB+ kanamycin + DAP. A 1:1 ratio of the donor and recipient CHS7 was mixed, pelleted, and resuspended in 100 µl of PBS. Suspensions were spotted on 0.45 µm filters and incubated on plates containing DAP at 37°C for 3 hr, after which the cells were resuspended in PBS and plated on LB+ kanamycin. Barcodes in 10 individual colonies were Sanger-sequenced to confirm that each recipient colony had a single distinct barcode. Finally, 1152 individual colonies were inoculated in twelve 96-well plates in LB+ kanamycin, grown overnight, pooled, and frozen in PBSG. Note that there is no cost to using more barcodes as it will improve the accuracy of STAMPR calculations. Future studies should employ as many barcodes as is reasonably possible.

## Barcode stability and influence on ExPEC growth

To assess if the integrated pDS132-STAMPR is maintained in the absence of selection, we cultured the library in the absence of kanamycin overnight. 1/1000 dilutions of the culture were serially passaged overnight for 5 days and plated daily on LB with or without kanamycin. Additionally, 30 colonies each day were patched from the plate lacking kanamycin onto LB+ kanamycin. All patched colonies on all 5 days were resistant to kanamycin, and thus possessed the barcode. To assess if the pDS132-STAMPR confers an in vitro growth defect, nine different barcoded strains along with CHS7 were growth overnight diluted 1/1000 and individually grown in triplicate in a 96-well plate overnight. OD600 was measured in a plate reader.

## Sample processing for sequencing

Bacterial samples obtained from either in vitro culture or organ homogenates were plated on LB+ kanamycin plates and incubated overnight at 37°C. If the cultures yielded a high density of cells, they were scraped and resuspended in PBSG and aliquots were stored at –80°C. If the cultures yielded few cells, each colony was individually picked and resuspended into the same PBSG solution (one tube per sample). Suspensions were frozen at –80°C until processing. Frozen aliquots of cells were later thawed and 2 µl was diluted in 100 µl of water. Samples were then boiled at 95°C for 15 min and 1 µl was used as template for PCR. Cycling conditions for PCR were the following: initial denaturation 98°C 30 s, 25 cycles of 98°C 15 s, 65°C 30 s, and 72°C 15 s, and final extension of 72°C 5 min. Reactions were carried out with a total volume of 50 µl using Phusion DNA polymerase (New England Biolabs). Amplified products were then run on 1% agarose gel to verify the presence of the amplicon, and all samples were pooled and purified using the GeneJet PCR Purification Kit (Fisher). Purified amplicons were quantified by Qubit and loaded on a MiSeq (Illumina) with either V2 or V3 reagent kits for 78 cycles. FASTQ files were then processed as described in *Hullahalli et al., 2021* to yield STAMPR measurements ($N_b$, $N_r$, GD, RD, FRD).

## Creation of the STAMPR calibration curve and barcode reference list

After the barcoded library was pooled, 100 µl of 10-fold serial dilutions were spread on LB+ kanamycin plates, corresponding to $10^1$ cells to $10^8$ cells. These samples were used to create a standard curve and serve as controls of known bottleneck sizes. All samples were prepared and sequenced as described above. Two replicates of the most undiluted sample were sequenced, and the dedupe plugin on Geneious (Biomatters) was used (k = 20, max substitutions = 2, max edits = 2) to deduplicate the reads to create a preliminary list of reference barcodes. Then, the two replicates used to create this list were combined with a third and mapped to the preliminary list of references in CLC Genomics Workbench (Qiagen) using the default mapping parameters. We then determined a depth threshold that corresponded to 97% of all reads. This was used to determine the set of 1329 barcodes that was used throughout this study. Note that this is slightly greater than the number of colonies picked (1152), thus providing a 'cushion' against errors in deduplication or mapping. This list of references was then used to map all samples of the calibration curve. The output vector containing the number of reads mapped to each barcode was then processed for data analysis as described in detail in *Hullahalli et al., 2021* to confirm that this list of references yields accurate FP estimates (*Figure 1—figure supplement 1C*).

## Creation of ExPEC transposon library

500 µl of *E. coli* donor MFD λ pir containing the transposon bearing plasmid pSC189 (*Chiang and Rubin, 2002*) grown overnight was mixed with 500 µl of CHS7 grown overnight. The mixture was pelleted, resuspended in 100 µl PBS and spotted on 0.45 µm filters and incubated overnight at 37°C. This procedure was performed in triplicate and tittered on LB+ kanamycin plates to quantify transconjugants. For the final library, ~7 million transconjugant colonies were obtained and pooled.

## In vivo ExPEC TIS protocol and reads processing

2.5 µl of the CHS7 transposon library was resuspended in 1 ml PBS, corresponding to 2E8 CFU/ml. 100 µl of this suspension was injected i.v. into five mice. The next day, the liver and spleens were harvested, homogenized, and plated on 245 mm (square) or 150 mm (circle) LB+ kanamycin plates, respectively. Cells were scraped in PBSG and stored in 1 ml aliquots. gDNA was then harvested using

the GeneJet gDNA Isolation Kit. Generation of libraries for sequencing was performed essentially as previously described (*Warr et al., 2019*). Briefly, 10 µg of gDNA was fragmented to 400 bp fragments with an ultrasonicator. Ends were then repaired with the Quick Blunting Kit (New England Biolabs) and A tailing was performed with Taq polymerase. Adapter sequences were ligated overnight with T4 DNA ligase. 29 cycles of PCR were then performed to enrich for transposon-containing sequencing, and an additional 19 cycles of PCR were performed to introduce sample indexes and Illumina sequences. 300–500 bp fragments were then gel excised, quantified, and loaded on a MiSeq for 78 cycles.

All steps to generate BAM mapping files were performed using CLC Genomics Workbench. First, FASTQ files were trimmed using the following parameters: 3′ sequence – ACCACGAC; 5′ sequence – CAACCTGT; mismatch cost = 1; gap cost = 1; minimum internal score = 7; minimum end score = 4; discard reads <10 nt. Trimmed reads were mapped to the CFT073 genome using the following parameters: mismatch score = 1; mismatch cost = 1; insertion cost = 3; deletion cost = 3; length fraction = 0.95; similarity fraction = 0.95. Reads that mapped to multiple locations were mapped randomly once, and a global alignment was performed to ensure accurate positions of terminal nucleotides. The resulting mapping file was exported in BAM format and processed as described below.

## RTISAn pipeline for TIS data analysis

We applied our new bottleneck calculation methodology (*Hullahalli et al., 2021*) to refine our previous TIS data analysis pipeline (Con-ARTIST) (*Pritchard et al., 2014*) as it did not sufficiently account for infection bottlenecks. This new R-based TIS analysis (RTISAn) employs the resampling strategy used here to calculate FP (*Hullahalli et al., 2021*) to accurately simulate infection bottlenecks from input TIS libraries. RTISAn serves as a successor to Con-ARTIST (*Pritchard et al., 2014*; *Hubbard et al., 2018*; *Hubbard et al., 2019*). The RTISAn approach is functionally similar to the 'resampling' method used in the TRANSIT pipeline but is implemented in R (*Subramaniyam et al., 2019*). The RTISAn pipeline begins with the user input of a FASTA file and locations of annotations or features. A 'TAinfo' file is generated that identifies the overlapping feature of each TA site. The TAinfo file is then used with a BAM mapping file to count the number of reads at each TA site, compiling this information in a 'TAtally' file.

The bulk of the RTISAn computation is used to compare two TAtally vectors, where one is an input and the other is an output. The goal is to identify genes for which insertions are either under- or over-represented in the output relative to the input. First, both the input and output vectors are normalized via a sliding window approach for replication bias. We set a window and step size of ~100,000 nt, so ~50 windows will be made for the *E. coli* genome. The mean of the bottom 99% of insertions is used to normalize each window, preventing the effects of clonal expansion from influencing the normalization.

Next, the normalized input vector must be resampled to match the saturation of the output vector. To accomplish this, the normalized input vector is iteratively resampled from a multivariate hypergeometric distribution 100 times, starting from 1000 reads and ending at the number of reads in the normalized input vector. The saturation is plotted against the sampling depth; the sampling depth that best approximates the saturation in the output vector is determined with inverse linear interpolation. The output represents a single sampling depth that accurately simulates the bottleneck on the input such that the saturation matches the output. To refine this estimate, we take a similar approach used to identify local minima in *Hullahalli et al., 2021*; instead of identifying local minima, we resample the input vector for 200 iterations at various sampling depths such that the resampling yields saturations closer to the saturation of the output vector than the previous sampling depth. In doing so, 200 sampling depths are acquired from which a mean and standard deviation are obtained. These values are used to sample a normal distribution 100 times. These 100 values represent read depths that when used to resample the input vector result in a saturation that approximates the saturation of the output vector. The final 100 simulations are performed at these read depths on the input vector.

Each simulation is compared on a gene-by-gene basis to the normalized output vector. We calculate a geometric mean Mann–Whitney p-value and mean fold change across all simulations. We also determine the number of 'informative sites,' defined as the number of TA sites that are nonzero across the output and all simulations. Genes with few informative sites in the input will usually not pass significance cutoffs but can still display large fold changes.

Since both p-value and fold change are separate informative metrics, a script that combines both sets of information to 'rank' genes when multiple animal replicates are used was created. This approach is partly unbiased, in that the user supplies a desired p-value and fold change cutoff for significance but the relative position of each gene with respect to the entire data set is considered as well. The rank is first determined by creating a score, from 0 to 1, that measures how many instances a gene passed either the p-value or fold change (with cutoffs of 0.05 and −1 in this study, respectively). For example, if five animals are used, then a score of 0.9 implies that, out of 10 metrics (five fold changes and five p-values), 9 passed the user-specified cutoffs. A separate unbiased rank is created by the mean rank of the average p-value and fold change for each gene. The final rank is then generated from the mean of the unbiased rank and the score. In practice, this strategy ensures that genes that consistently pass the user-specified cutoff will always be ranked above genes that pass the cutoffs fewer times; this latter category can then be subdivided into the relative magnitude of fold change and p-value.

The entire RTISAn pipeline is available at https://github.com/hullahalli/stampr_rtisan (copy archived at swh:1:rev:d6eadda53c1ef7a78cebbebd09a2243674351091, *Hullahalli and Waldor, 2021*) including TAtally files to reproduce the comparisons in this study.

## Analysis of barcoded mutant libraries in vivo

Barcoding of each mutant was performed by conjugation of the same donor library used to generate the CHS7 barcoded library. Three colonies per mutant were Sanger-sequenced to establish the barcode identity, cultured overnight, and pooled to generate a mixed library of 21 tags in seven strains (*mlaC, yejM, ftsX, manA, nagA, nagK*, and CHS7). This pool was diluted in PBS and 100 µl was injected i.v. into mice. 100 µl was also directly plated on LB+ kanamycin plates to serve as the input comparison. The bile, spleen, lungs, right kidney, and all four lobes of the liver (separately) were harvested and plated on 150 mm LB+ kanamycin plates. After overnight incubation at 37°C, cells were scraped and processed as described above. Sequencing reads were mapped to the list of 21 barcodes and custom R scripts were used to quantify the change in barcode frequency relative to the input. Note that the method to generate barcoded libraries is the same as previously described (*Warr et al., 2019*; *Hubbard et al., 2016*), although the analytical pipeline differs due to the necessity of visualizing clonal expansion.

## Statistical analysis

p-Values reported in figures were obtained from GraphPad Prism 9. Figure legends indicate the statistical test used.

## Acknowledgements

This work was supported by an NSF Graduate Research Fellowship (KH) and NIH R01 AI-042347 and Howard Hughes Medical Institute (MKW). We are grateful to Bolutife Fakoya, Abdelrahim Zoued, Carole Kuehl, Ting Zhang, and Ian Campbell for assistance with plating organ homogenates. We are grateful to members of our lab and Michael Chao for providing feedback on the manuscript.

## Additional information

### Funding

| Funder | Grant reference number | Author |
| --- | --- | --- |
| Howard Hughes Medical Institute | | Matthew K Waldor |
| National Science Foundation | | Karthik Hullahalli |
| National Institutes of Health | 1 F31 AI156949-01 | Karthik Hullahalli |

| Funder | Grant reference number | Author |
|---|---|---|
| National Institutes of Health | R01 AI-042347 | Matthew K Waldor |

The funders had no role in study design, data collection and interpretation, or the decision to submit the work for publication.

### Author contributions
Karthik Hullahalli, Conceptualization, Data curation, Formal analysis, Investigation, Methodology, Project administration, Resources, Software, Validation, Visualization, Writing - original draft, Writing - review and editing; Matthew K Waldor, Conceptualization, Funding acquisition, Project administration, Resources, Visualization, Writing - original draft, Writing - review and editing

### Author ORCIDs
Karthik Hullahalli (ID) http://orcid.org/0000-0003-3064-2090
Matthew K Waldor (ID) http://orcid.org/0000-0003-1843-7000

### Ethics
All animal experiments were conducted in accordance with the recommendations in the Guide for the Care and Use of Laboratory Animals of the National Institutes of Health and the Animal Welfare Act of the United States Department of Agriculture using protocols reviewed and approved by Brigham and Women's Hospital Committee on Animals (Institutional Animal Care and Use Committee protocol number 2016N000416 and Animal Welfare Assurance of Compliance number A4752-01).

### Decision letter and Author response
Decision letter https://doi.org/10.7554/eLife.70910.sa1
Author response https://doi.org/10.7554/eLife.70910.sa2

## Additional files

### Supplementary files
- Supplementary file 1. Genes identified by transposon insertion sequencing (TIS).
- Supplementary file 2. Strains and plasmids.
- Supplementary file 3. Oligonucleotides.
- Transparent reporting form

### Data availability
Barcode and TIS read counts, in addition to all scripts required to analyze these data, are available at https://github.com/hullahalli/stampr_rtisan (copy archived at https://archive.softwareheritage.org/swh:1:rev:d6eadda53c1ef7a78cebbebd09a2243674351091). Sequencing data for the time-resolved STAMPR (PRJNA757769), dosing (PRJNA757737), clodronate (PRJNA758036), TIS (PRJNA757442), and barcoded mutant (PRJNA757753) experiments have been deposited in the Sequencing Read Archive.

The following dataset was generated:

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
