## [Decision Letter]

**Acceptance summary:**

Monitoring changes in pathogenic bacterial populations after bloodstream infection are poorly understood and in this work the authors develop new genetic analytic tools to dissect bacterial population dynamics at sites of infection. By harvesting bacteria from different sites and times of infection and performing deep sequencing to define the distribution of specific tagged-strains the authors provide a highly detailed snapshot of populations, with discriminatory power superior to prior studies. The work paves the way to future such studies in other organisms that cause persistent infection in humans.

**Decision letter after peer review:**

Thank you for submitting your article "Pathogen clonal expansion underlies multiorgan dissemination and organ-specific outcomes during systemic infection" for consideration by *eLife*. Your article has been reviewed by 3 peer reviewers, and the evaluation has been overseen by a Reviewing Editor and Bavesh Kana as the Senior Editor. The reviewers have opted to remain anonymous.

Essential revisions:

1. Add a conclusion sentence at the end of abstract. In both the abstract and introduction, a clear articulation of the various objectives of the study would be helpful to orient readers. The abstract would also benefit from presenting specific results, rather than simply a descriptive summary. In particular, it would be useful to mention both hexose metabolism and cell surface gene contributions in the abstract, as summarized in the introduction.

2. Line 77-78: The ref is a preprint. To help the readers to understand the model, a short description how these parameters were calculated may be added in the corresponding sections.

3. Line 114: Please comment on how the numbers of barcodes will affect the results. In other words, if different barcodes will be used, will similar results be obtained? Were there recommended barcode numbers for such an experiment?

4. Line 154-158: Were there any mutational changes that can explain the high CFU burden among the "outliners"? Were any mutations found in the genes involved in the cell envelope hemostasis or fructose-6-phosphate metabolism (e.g. line 346, 347)?

5. There appears to be a disconnection between the STAMPR and TnSeq analysis, and it would be interesting to see the correlations or some discussion.

6. Line 588: Please specify if each colony represents a unique barcode.

7. Line 603-605, 611: It is unclear why individual colonies were analyzed separately. How were the amplicons pooled? Equal amounts? The steps may introduce variations for deep sequencing analysis.

8. The authors show that changes in different parameters (i.e. inoculum) impact the trajectories followed by the pathogens. One can imagine that in a normal scenario, the number of bacterial cells that will arrive to the blood will be much lower than the inoculum used in these experiments. Similarly, host factors will also influence which people will be infected or not. The authors should explain more clearly why the model used is relevant to understand natural infections.

9. The authors clearly show that *E. coli* CFT073 colonizes the liver much better than other organs. Is there any evidence that liver infections are over-represented in patients suffering with *E. coli* bacteremia?

10. In the last part of the paper, the authors identified several genes involved in pathogen survival after infection (early stages). Was this clonal expansion a stochastic process, not driven by mutations in specific genes? Can the authors sequence some of the clones that expanded to clearly show if this process of clonal expansion was random (or conversely, was driven in some cases by mutations in specific genes)?

11. Please elaborate a bit on what is meant by "distinct cellular niches" in the lungs, and how these might account for organ specific results.

12. It is unclear why the hexose metabolism findings are surprising, as carbon metabolism and alternative carbon pathways are implicated in phagocytosis and phagosome resistance in various bacteria and yeast like Candida.

13. There is an interesting paper from Fitzgerald's group (PMID: 31807698), analyzing both intra and inter-host dissemination that should be referenced here.

14. In addition to papers cited in the discussion that this study corroborates and expands upon, it would be appropriate to cite older literature of the "single organism" hypothesis by Rubin and others in the 1980s and earlier.

15. Line 730-732: The raw reads should be made accessible to the community.

*Reviewer #1 (Recommendations for the authors):*

Abstract: Add a conclusion sentence at the end of abstract.

Line 77-78: The ref is a preprint. To help the readers to understand the model, a short description how these parameters were calculated may be added in the corresponding sections.

Line 114: Please comment on how the numbers of barcodes will affect the results. In other words, if different barcodes will be used, will similar results be obtained? Were there recommended barcode numbers for such an experiment?

Line 154-158: Were there any mutational changes that can explain the high CFU burden among the "outliners"? Were any mutations found in the genes involved in the cell envelope hemostasis or fructose-6-phosphate metabolism (e.g. line 346, 347)?

There appears to be a disconnection between the STAMPR and TnSeq analysis, and it would be interesting to see the correlations.

Line 588: Please specify if each colony represents a unique barcode.

Line 603-605, 611: Not sure why individual colonies were analyzed separately. How were the amplicons pooled? Equal amount? The steps may introduce variations for deep sequencing analysis.

Line 730-732: The raw reads should be made accessible.

*Reviewer #3 (Recommendations for the authors):*

In both the abstract and introduction, a clear articulation of the various objectives of the study would be helpful to orient readers.

The abstract would also benefit from presenting specific results, rather than simply a descriptive summary. In particular, I think it would be useful to mention both hexose metabolism and cell surface gene contributions in the abstract, as summarized in the introduction.

In addition to papers cited in the discussion that this study corroborates and expands upon, it would be appropriate to cite older literature of the "single organism" hypothesis by Rubin and others in the 1980s and earlier.

Did the authors do any sequencing to assure that unintentional mutations weren't introduced to the *E. coli* strain, beyond simply Sanger sequencing of tags? How do the authors know that strains were in fact clonal and isogenic?

Please elaborate a bit on what you mean by "distinct cellular niches" in the lungs, and how these might account for organ specific results.

Not sure why the hexose metabolism findings are surprising, as carbon metabolism and alternative carbon pathways are implicated in phagocytosis and phagosome resistance in various bacteria and yeast like Candida.

---

## [Author Response]

Essential revisions:1. Add a conclusion sentence at the end of abstract. In both the abstract and introduction, a clear articulation of the various objectives of the study would be helpful to orient readers. The abstract would also benefit from presenting specific results, rather than simply a descriptive summary. In particular, it would be useful to mention both hexose metabolism and cell surface gene contributions in the abstract, as summarized in the introduction.

A conclusion was added to the abstract (lines 44-46).

A clear articulation of the key objective of the study was added to the abstract (line 32-35). In our view, the introduction also provides an overview of the context and objectives of the study in better detail.

Specific data regarding hexose metabolism and cell surface homeostasis genes are now mentioned in the abstract (42-44).

2. Line 77-78: The ref is a preprint. To help the readers to understand the model, a short description how these parameters were calculated may be added in the corresponding sections.

This reference is no longer a preprint and is updated in the reference list. We have included descriptions of the parameters in the Results section prior to discussing them in the context of our system (lines 135-144 and 217-234).

3. Line 114: Please comment on how the numbers of barcodes will affect the results. In other words, if different barcodes will be used, will similar results be obtained? Were there recommended barcode numbers for such an experiment?

Our companion manuscript goes into greater detail regarding the impact of barcode number on STAMPR resolution. When more barcodes are used, the upper resolution limit increases, generally increasing the precision of the analysis. Thus the recommended number of barcodes is as high as can be feasibly obtained (line 629).

4. Line 154-158: Were there any mutational changes that can explain the high CFU burden among the "outliners"? Were any mutations found in the genes involved in the cell envelope hemostasis or fructose-6-phosphate metabolism (e.g. line 346, 347)?

Whole-genome sequencing was not carried out on individual clones in this study. However, since expanded clones comprised different barcodes across different mice and across independent creations of the barcode library, it is extremely unlikely that potential causative mutants were pre-existing in the library. It is also unlikely that mutations occurred in the animal prior to expansion, given that expansion is observed as early as 1 dpi when relatively few doublings have occurred. Furthermore, the discrete localization of liver abscesses would require that causative mutations occurred in different bacterial cells that were somehow also clustered together in physical space, which is extremely unlikely. We have expanded these points in the revised discussion (lines 531-544).

However, these points do not definitively disprove that mutations are involved in the expansion process. Whole genome sequencing individual clones would almost certainly identify mutations, but only secondary phenotypic validation will confirm that these mutations are linked to expansion. We plan to isolate clones within abscesses and reinfect mice with these clones. However, this experiment is beyond the scope of the present study.

5. There appears to be a disconnection between the STAMPR and TnSeq analysis, and it would be interesting to see the correlations or some discussion.

These sections are admittedly more disconnected than the rest of the manuscript, but there are important and practical parallels between STAMPR and TIS:

1. TIS is inherently limited by bottlenecks, which causes mutants to randomly drop out of the population by chance rather than through selection. Identifying the conditions (e.g., organs, time points, and replication) with the widest bottlenecks via STAMPR is ideally suited to optimize TIS experiments.

2. Since TIS is genome-scale, the rate of false positives is much higher than competitions between individual mutants. However, screening fitness determinants individually without barcodes is laborious. Barcoding mutants, however, is an approach that balances the tradeoff between false positives and throughput; multiple mutants can be validated simultaneously without false positives. Since a single PCR reaction can be used to simultaneously evaluate the fitness of several mutants, barcoding also enables rapid multiorgan and temporal analyses.

3. STAMPR (and barcodes more broadly) helps contextualize genetic determinants in the broader population-level landscape. For example, hexose metabolism is specific in both temporal and spatial contexts, whereas cell envelope homeostasis appears to be a more ubiquitous fitness factor in the host.

4. Since TIS requires the use of mutant libraries, STAMPR acts complementarily with isogenic libraries that can discern which population-level patterns are unlikely to be driven by genotypes pre-existing in the library.

5. STAMPR can identify discrete patterns of persistence and expansion, and TIS helps narrow down a set of bacterial pathways that could further distinguish these population-level phenomena.

We have included these points throughout the manuscript and more directly in Lines 493-497

6. Line 588: Please specify if each colony represents a unique barcode.

Each colony represents a unique barcode at this FP range (Line 627)

7. Line 603-605, 611: It is unclear why individual colonies were analyzed separately. How were the amplicons pooled? Equal amounts? The steps may introduce variations for deep sequencing analysis.

We apologize for the confusion. Individual colonies were not analyzed separately, but individually picked and pooled together (line 646). This is purely for technical reasons – flooding a plate with 1-5 ml of PBSG when there are very few colonies will result in a relatively dilute bacterial suspension. Individually picking and pooling colonies into a small volume of PBSG is a simpler way of creating a denser bacterial suspension.

Amplicons were pooled in approximately equal amounts, as determined by band intensity on a gel. Differences in band intensities have a negligible impact on STAMPR measurements, since samples that amplify less efficiently were derived from a more diluted bacterial suspension, with often as little as 1 colony. Because the FP of these samples are so low, they require fewer reads to make accurate measurements.

8. The authors show that changes in different parameters (i.e. inoculum) impact the trajectories followed by the pathogens. One can imagine that in a normal scenario, the number of bacterial cells that will arrive to the blood will be much lower than the inoculum used in these experiments. Similarly, host factors will also influence which people will be infected or not. The authors should explain more clearly why the model used is relevant to understand natural infections.

This is an excellent point, and a recent review by the Mobley and Bachman labs addresses some of these points in detail was published recently. We have included a paragraph centered around these points in the discussion [lines 425-463].

9. The authors clearly show that *E. coli* CFT073 colonizes the liver much better than other organs. Is there any evidence that liver infections are over-represented in patients suffering with *E. coli* bacteremia?

*E. coli* are among the leading causes of liver abscesses, and these infections are often associated with bacteremia (line 453). We were unable to find evidence of the reverse (liver infections in patients with bacteremia).

10. In the last part of the paper, the authors identified several genes involved in pathogen survival after infection (early stages). Was this clonal expansion a stochastic process, not driven by mutations in specific genes? Can the authors sequence some of the clones that expanded to clearly show if this process of clonal expansion was random (or conversely, was driven in some cases by mutations in specific genes)?

See response to comment #4. It is highly unlikely that there is a genotypic change that permits clonal expansion, but as noted by reviewer 2, full investigation of the nature of expansion will require a future study. Although clonal expansion is not driven by mutations, there are likely bacterial genes that are important for expansion (or persistence), and the TIS was performed to help identify these genes.

11. Please elaborate a bit on what is meant by "distinct cellular niches" in the lungs, and how these might account for organ specific results.

Done (455-456)

12. It is unclear why the hexose metabolism findings are surprising, as carbon metabolism and alternative carbon pathways are implicated in phagocytosis and phagosome resistance in various bacteria and yeast like Candida.

The surprise was primarily that these genes were organ specific and varied in importance across time. The language has been honed to clarify this point (line 353).

13. There is an interesting paper from Fitzgerald's group (PMID: 31807698), analyzing both intra and inter-host dissemination that should be referenced here.

We have included the reference (line 544)

14. In addition to papers cited in the discussion that this study corroborates and expands upon, it would be appropriate to cite older literature of the "single organism" hypothesis by Rubin and others in the 1980s and earlier.

Thank you, we now cite these early papers, which are relevant indeed (line 427-438).

15. Line 730-732: The raw reads should be made accessible to the community.

Sequencing reads have been uploaded to SRA (lines 773-775).